# Identifying irrigated areas using land surface temperature and hydrological modelling: application to the Rhine basin

**Devi Purnamasari**[1,2]**, Adriaan J. Teuling**[1]**, and Albrecht H. Weerts**[1,2]

[1]Hydrology and Environmental Hydraulics Group, Wageningen University, Wageningen, the Netherlands
[2]Deltares, Delft, the Netherlands

**Correspondence:** Devi Purnamasari (devi.purnamasari@deltares.nl)

**Abstract.** Information about irrigation with relevant spatiotemporal resolution for understanding and modelling irrigation dynamics is important for improved water resource management. However, achieving a frequent and consistent characterization of areas where signals from rain-fed pixels overlap with irrigated pixels has been challenging. Here, we identify irrigated areas using a novel framework that combines hydrological modelling and satellite observations of land surface temperature (LST). We tested the proposed methodology on the Rhine basin covering the period from 2010 to 2019 at a 1 km resolution. The result includes multiyear irrigated maps and irrigation frequency. Temporal analysis reveals that an average of 159 000 ha received irrigation at least once during the study period. The proposed methodology can approximate irrigated areas with $R^2$ values of 0.79 and 0.77 for 2013 and 2016 compared to irrigation statistics, respectively. In dry regions, the method performs slightly better than in wet regions with $R^2$ values of 0.90 and 0.87 in respective years, with an average improvement in $R^2$ by 0.14. The method approximates irrigated areas in regions with large agricultural holdings better than in regions with small fragmented agricultural holdings, due to binary classification and the choice of spatial resolution. The irrigated areas are mainly identified in the established areas indicated in the existing irrigation maps. A comparison with global datasets reveals different disparities due to spatial resolution, input data, reference period, and processing techniques. From the multiyear results, the largest irrigated area was found in the Alsace region in the Rhine valley, where the irrigation extent is negatively correlated with precipitation ($r = -0.82$, $p$ value $= 0.004$) and less with potential evapotranspiration (ET).

## 1 Introduction

The expansion of irrigated areas, resulting from the concurrent effects of a growing population and climate change, continues to exert pressure on water resources (Döll and Siebert, 2002). In warmer and drier climates, an increase in crop evapotranspiration (ET) is expected to increase net irrigation water to maintain or improve agricultural yields (Fader et al., 2016; Fischer et al., 2007). However, future water availability will be negatively affected by changes in temperature and precipitation, raising concerns about whether there will be enough water to meet the growing demand (Konapala et al., 2020; Boretti and Rosa, 2019). Recent summer drought events in Europe have been exceptional, characterized by widespread soil moisture deficits and a significant decrease in water resource availability (Spinoni et al., 2018; Hanel et al., 2018). These events have had a profound impact on agriculture, with reported yield in 2018 surpassing 50 % compared to the average yield of the previous 5 years (Toreti et al., 2019). In the future, farmers may increasingly turn to irrigation to mitigate crop losses that can create conflicts with other water users. The Rhine basin serves as an example, being one of the major northern humid rivers affected by recent extreme droughts through its sensitivity to evapotranspiration (Buitink et al., 2021). It experienced extremely low water levels in consecutive summer months of 2018–2019 that caused water supply bottlenecks and disruptions in inland navigation in Germany (BfG, 2019). While past drought events have been studied in terms of their frequency and severity, there remains limited understanding on how irrigation intensifies pressure on water resources.

Identifying where irrigation occurs and how it evolves over time can offer improved insight into water use for sustainable water resource planning and management. Unfortunately, maps with irrigation extent with relevant spatial and temporal resolution for water management at the basin level are often lacking. This results in challenges in estimating irrigation water requirements and developing hydrological models. Most research efforts have focused on monitoring the spatiotemporal extent of irrigated areas and quantifying irrigation rates in arid and semi-arid climates (see the Murray–Darling Basin (Peña-Arancibia et al., 2016), the Ebro Basin (Dari et al., 2021, 2023; Zappa et al., 2024; Kragh et al., 2024), and the Miandoab plain in Iran (Jalilvand et al., 2019)). For the Rhine basin, the primary source of information on irrigated areas comes from sub-national statistics which are data sources for developing previous global maps of irrigated areas, such as the Global Map of Irrigation Areas (GMIA) (Siebert et al., 2005) and MIRCA2000 (Portmann et al., 2010). There is an increasing need to expand these research efforts for better-informed decisions in water resource management. In humid and temperate regions, shifting climatic conditions may offer advantages to the agricultural sector as larger areas become more suitable for crop cultivation, leading to a potential increase in irrigation water demands (Iglesias et al., 2012).

Researchers have used vegetation indices, such as the normalized difference vegetation index (NDVI) and the enhanced vegetation index (EVI), derived from optical sensors to detect irrigated areas in large regions (Xie et al., 2021; Bretreger et al., 2020; Abera et al., 2021). These indices typically capture vegetation health and growth stages, with irrigated fields exhibiting higher values than adjacent non-irrigated fields. However, most studies are performed in areas with negligible precipitation during the growing season, where spectral difference is more pronounced. In temperate regions, distinguishing between irrigated and non-irrigated croplands using vegetation indices is challenging, as irrigation often supplements precipitation, which leads to overlap in the spectral signatures of irrigated and non-irrigated areas. A study by Shamal and Weatherhead (2014) revealed that the spectral signatures between irrigated and non-irrigated croplands in the UK were identical because non-irrigated croplands experienced less water stress due to regular precipitation. Similar findings were reported by Ozdogan and Gutman (2008), who attempted to identify irrigated areas in the US, but the performance results deteriorated when applied to the humid eastern regions. Previous studies suggest including additional information, such as climatic information, land-use maps, and other remote sensing datasets, to improve the identification of irrigated fields (Peña-Arancibia et al., 2016; Deines et al., 2019; Ozdogan and Gutman, 2008).

One of the land variables affected by vegetation water stress is land surface temperature (LST). During water-limited conditions, reduced evapotranspiration increases LST and drives an increase in sensible heat flux. In contrast, irrigated areas generally show lower LST compared to non-irrigated croplands. The use of LST as an indicator of crop health resulting from irrigation has been applied in arid and semi-arid regions. Zhu and Burney (2022) highlighted the effectiveness of using LST observations in crop models to quantify evaporative cooling effects from changes in water and surface energy over irrigated maize croplands in Nebraska in the United States. Their findings demonstrate that LST shows the impacts of irrigation on heat and water stress in crops. Olivera-Guerra et al. (2020) used LST as complementary data in crop models to estimate irrigation water use. By comparing elevated LST with the canopy temperature of well-watered fields, they were able to quantify the crop water stress coefficient ($K_s$). Haddeland et al. (2006) investigated the impact of irrigation on the water and energy balances in the Colorado River and Mekong River basins using the variable infiltration capacity (VIC) hydrology model. The results show that, on an annual scale, the cooling effect from increased latent heat flux averaged 0.04 °C in both basins, with a more significant decrease of up to 2.1 °C during peak irrigation months in regions with dense irrigated croplands.

Although LST provides a clear difference between irrigated and rain-fed croplands in arid and semi-arid regions, its effectiveness diminishes in energy-limited conditions such as in temperate and humid climates. In regions with low surface energy availability, the use of LST is more challenging due to high moisture levels, reduced temperature variability, and the overlap of wet and dry periods, which complicate the separation of irrigation effects from natural variations in soil moisture and temperature (Roth et al., 2013). Zhang et al. (2022) used LST to estimate evapotranspiration from irrigation in the North China Plain, achieving higher accuracy in winter, when precipitation is lower. During summer months, the effects of irrigation on LST are more difficult to detect, as precipitation often meets crop water needs, making irrigation supplemental and its impact on LST minimal. In such conditions, complementary methods are required for accurate irrigation detection. The more stable moisture levels and less pronounced temperature fluctuations make it difficult to differentiate between irrigated and non-irrigated areas based solely on LST.

To improve irrigation detection, we exclude precipitation-driven evapotranspiration estimated by the wflow_sbm hydrological model (van Verseveld et al., 2024) from evapotranspiration driven by irrigation to provide more distinct features for classification. We integrated surface energy into the water balance by linking evapotranspiration to land surface temperature, as irrigation water use accounts for a significant portion of consumptive water loss in the form of actual evapotranspiration, which is governed by climatic conditions (Peña-Arancibia et al., 2016; Droogers et al., 2010). Existing approaches often involve comparing satellite-based retrievals with estimated ET fluxes derived from hydrological models (Velpuri and Senay, 2017; Romaguera et al., 2012). However, the accuracy of satellite-based evapotranspiration

retrievals depends on how well the partitioning of evapotranspiration is modelled, which is still largely unvalidated (Talsma et al., 2018; Wang and Dickinson, 2012). Additionally, ET estimates from remote sensing models are highly divergent across products, with inconsistencies attributed to differences in input data, methodology, parameterization, and model structure (Vinukollu et al., 2011; Badgley et al., 2015; Lehmann et al., 2022). Zhang et al. (2023) elaborated on the significant divergence between ET estimates from energy balance approaches and residual water balance methods in humid regions. Although ET models capture monthly variations, they show different sensitivities to precipitation and often fail to capture the spatial patterns of ET from water balance methods and the variability caused by ET peaks following heavy precipitation. It is argued that minimizing ET errors can be achieved by ensuring proper partitioning of the water balance, constraining the magnitude of precipitation, and selecting high-quality datasets (Lehmann et al., 2022).

This paper investigates the potential of using a framework that combines evapotranspiration estimates from a spatially distributed hydrological model wflow_sbm (van Verseveld et al., 2024) and the MODIS LST product to detect and monitor irrigated areas. We use an additional surface energy balance module that links evapotranspiration estimates to LST, enabling direct comparison with satellite observations. Our research aims to address the following questions based on the outcomes of this study:

1. Could the difference in land surface temperature between simulated values from evapotranspiration estimates from the wflow_sbm model and satellite observations identify irrigated areas when compared against available regional statistics of irrigated areas or existing irrigated maps?

2. What is the extent of the irrigated areas around the Rhine, and what controls its interannual variability?

## 2 Data and methodology

### 2.1 Study area

We tested the proposed methodology to identify irrigated areas in the Rhine basin as shown in Fig. 1. It drains an area of approximately $160\,000\,\text{km}^2$. Figure 1b shows land uses and land cover in the basin, where agriculture occupies approximately 46 % of the total land use according to Copernicus CORINE Land Cover (CLC 2018) data (European Environment Agency, 2018). The agricultural fields are characterized by the cultivation of various crops, including cereals, oilseeds, potatoes, and sugar beets. A notable feature of this agricultural landscape is the prevalence of irrigation systems in the Middle Rhine basin, which stretches from south to north along the border between France and Germany. Supplementary irrigation is commonly practised during the summer months to prevent agricultural loss. Sources of irrigation come primarily from surface water bodies, groundwater bodies, reclaimed wastewater, and rainwater collection. Based on the EU Water Framework Directive (2000/60/EC) (WFD), each EU member state is required to regulate water abstraction through prior authorization regimes and provide incentives for efficient water use. For example, France has introduced taxation and mandatory metering as economic instruments related to surface and groundwater abstraction (Berbel et al., 2019).

Precipitation and potential evapotranspiration play important roles in determining water availability and demand for irrigation. Table 1 summarizes the mean seasonal precipitation and potential evapotranspiration in the Rhine basin for 2010–2019. The Middle Rhine and East Alpine sub-basins are representative of the two main seasonal cycles in the basin. The East Alpine region had higher precipitation than potential evapotranspiration compared to other sub-basins, while the Middle Rhine had relatively similar annual precipitation and evapotranspiration rates. However, the evapotranspiration rate in spring (MAM) and summer (JJA) often surpasses the precipitation, reflecting the potential for a water-limited regime. These fluctuations in precipitation and evapotranspiration throughout the year can influence the extent of irrigated areas annually. However, publicly accessible data regarding multiyear irrigated maps of the Rhine basin are currently unavailable. The available information at the sub-national level (NUTS 2 unit) as shown in Fig. 1c, compiled by Eurostat, primarily relies on summaries derived from the Farm Structure Surveys (FSSs) conducted by EU member states. To identify irrigated areas within the Rhine basin, training and test data for supervised classification were collected from regions where irrigated plots can be identified through remote sensing observations, as described in Sect. 2.4.2.

### 2.2 Daily $ET_a$ from wflow_sbm

The wflow_sbm (van Verseveld et al., 2024) is a spatially distributed hydrological model designed to solve hydrological processes numerically at the grid cell. It accounts for several key hydrological processes: (1) canopy interception, (2) snow and glaciers, (3) soil moisture module and evapotranspiration, (4) lateral subsurface flow, (5) surface routing, and (6) reservoirs and lakes. The model takes both vertical and lateral processes into account when partitioning precipitation into storage, drainage, and evapotranspiration. Vertical processes are conceptualized as a soil bucket with saturated and unsaturated storage similar to Topog_SBM (Vertessy and Elsenbeer, 1999), while the lateral components (surface and subsurface flows) are routed using the kinematic-wave approximation. In this study, our focus lies on the evapotranspiration estimates of wflow_sbm due to its association with the land surface energy balance. The following gridded datasets, provided in daily temporal resolution and with

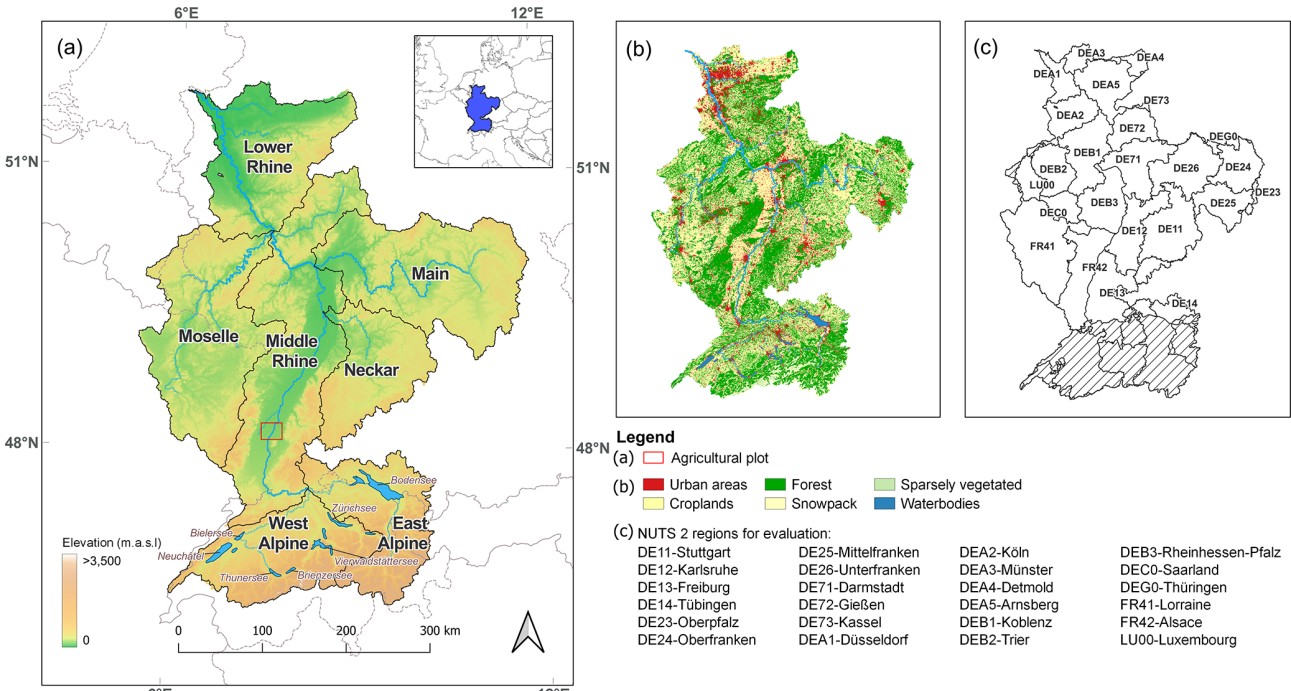

**Figure 1.** Overview of the Rhine basin: **(a)** the sub-basins from HydroSHEDS (Lehner et al., 2008) and a digital elevation model (Farr et al., 2007), **(b)** aggregated land use and land cover from the CORINE Land Cover 2018 (European Environment Agency, 2018), and **(c)** NUTS level 2 regions for which the reported total irrigated area was used to evaluate the results of classification analysis. The demarcated red line in panel **(a)** shows one of the croplands used to collect training and test data for building the supervised classification model. Hashed regions indicate areas where irrigated area data are not available.

**Table 1.** The mean seasonal (DJF, MAM, JJA, and SON) precipitation and potential evapotranspiration of the Rhine sub-basins from 2010–2019.

| Sub-basins | Precipitation (mm yr$^{-1}$) | | | | Potential evapotranspiration (mm yr$^{-1}$) | | | |
|---|---|---|---|---|---|---|---|---|
| | DJF | MAM | JJA | SON | DJF | MAM | JJA | SON |
| Middle Rhine | 235 | 198 | 240 | 238 | 59 | 220 | 320 | 70 |
| East Alpine | 286 | 314 | 448 | 307 | 62 | 222 | 313 | 75 |

a spatial resolution of 1 km, were used to compute water balance in wflow_sbm. According to a previous study by Imhoff et al. (2020), the choice of a 1 km spatial resolution is deemed sufficient to capture hydrological processes at the river basin level given the availability of data for hydrological parameters.

1. The precipitation data are obtained from the HYRAS dataset, which was developed by the German Meteorological Service (DWD) and the Federal Institute of Hydrology (BfG) (Rauthe et al., 2013).

2. The mean air temperature was derived from interpolating ground measurements with topographic correction based on the lapse rate (Van Osnabrugge et al., 2019).

3. Potential evapotranspiration was estimated based on the Makkink equation using ground observations of mean air temperature and downward shortwave radiation estimates from satellite products (Van Osnabrugge et al., 2019).

Evapotranspiration in wflow_sbm is expressed as a fraction of potential evapotranspiration that changes according to the amount of available water in the rooting zone (Feddes et al., 1976). Thus, the spatial variations in evapotranspiration across different land uses inherently vary depending on the rooting depth of vegetation, which can be inferred from information provided by the soil map. The model represents the soil as a column with several layers, allowing it to account for vertical water movement and variations in soil moisture. The movement of water in the unsaturated soil layer follows the Brooks–Corey model, which relates to the vertical saturated hydraulic conductivity and soil matrix potential. The rate of soil evaporation from unsaturated soil layers varies ac-

cording to the fraction of vegetation roots and the soil moisture content that is related to the soil water holding capacity. Therefore, the representation of the soil water holding capacity is crucial for estimating soil moisture and consequently evapotranspiration in the wflow_sbm model.

In humid regions, when precipitation exceeds potential evapotranspiration, excess precipitation tends to contribute to runoff rather than additional ET. To account for this process, the hydrological model needs to be calibrated and validated to perform well under rain-fed conditions. Additionally, this ensures that LST-derived ET estimates are constrained by potential evapotranspiration and that excess precipitation is accurately routed into runoff. Here, we use the most recent wflow_sbm schematization and parameterization as developed for the Dutch Ministry of Infrastructure and Waterways (see the report by Buitink et al., 2023). More detailed information on the parameterization, calibration, and validation of the wflow_sbm model is provided in Imhoff et al. (2020) and Eilander et al. (2021). The performance of the water balance model used in this study was validated against discharge measurements from various stations in the study basin, resulting in Kling–Gupta efficiency (KGE) coefficients ranging from 0.60 to 0.90 (Imhoff et al., 2020). It is important to note that wflow_sbm does not incorporate land management practices, such as irrigation, which could potentially lead to an underestimation or overestimation of actual evapotranspiration.

## 2.3 Land surface temperature module

The aim of this study is to determine the spatiotemporal pattern of irrigated areas by using the land surface temperature difference ($\Delta T_s$) as outlined in Fig. 2. This difference is obtained by comparing the land surface temperature derived from evapotranspiration ($ET_a$) estimates obtained from the wflow_sbm model ($T_{s,sim}$) with those obtained from satellite observations ($T_{s,obs}$). To achieve this, we have developed a module that connects the partitioning of surface energy balance fluxes with evapotranspiration estimates. This additional module is based on a parsimonious model previously coupled with the mesoscale hydrologic model (mHM) developed by Zink et al. (2018). Daily land surface temperature is derived from the sensible heat flux ($H$, $\mathrm{W\,m^{-2}}$), where it is obtained by resolving the energy balance equation, which requires the net available surface energy ($R_n$, $\mathrm{W\,m^{-2}}$), latent heat flux ($LE$, $\mathrm{W\,m^{-2}}$), and soil heat flux ($G$, $\mathrm{W\,m^{-2}}$) at a daily temporal resolution. The energy balance of the land surface is calculated as follows:

$$R_n = LE + H + G. \tag{1}$$

As the magnitude of the daytime $G$ is relatively small compared to $R_n$, the energy balance equation is expressed as follows:

$$H \approx R_n - LE. \tag{2}$$

The evapotranspiration ($\mathrm{mm\,d^{-1}}$), which is the water balance term provided by the wflow_sbm, is converted to latent heat flux $LE$ in the following:

$$LE = \lambda \times \rho_{\mathrm{water}} \times ET, \tag{3}$$

where $\rho_{\mathrm{water}}$ is $1000\,\mathrm{kg\,m^{-3}}$ and $\lambda$ is the latent heat vaporization ($\mathrm{J\,kg^{-1}}$). After obtaining sensible heat flux from Eq. (2), the land surface temperature $T_s$ can be computed as follows:

$$T_s = \frac{H r_a}{\rho_a c_p} + T_a, \tag{4}$$

where $r_a$ is aerodynamic conductance ($\mathrm{s\,mm^{-1}}$), $c_p$ is the specific heat of air ($\mathrm{J\,kg^{-1}\,K^{-1}}$), and $\rho_a$ is the density of air ($\mathrm{kg\,m^{-3}}$). The detailed equations used in this step are provided in Appendix A.

In summary, the land surface temperature $T_s$ is calculated using the following inputs: $R_s^{in}$ and $\alpha$ are obtained from satellite observations, $T_a$ represents the mean air temperature and serves as an input for the wflow_sbm model, $LE$ is derived from the evapotranspiration calculated by the wflow_sbm, and $r_a$ represents the aerodynamic resistance. The proposed land surface temperature module requires additional radiative terms as input. For this study, data from the geostationary satellites Meteosat Second Generation (MSG), the downward shortwave radiation (LSA-SAF DSSF) and surface albedo (LSA-SAF AL) at a spatial resolution of 3 km (Trigo et al., 2011), were used as radiative input data. As there is limited availability of daily land surface albedo data since 2009, we use land surface albedo from every 10th day available from 2005 onwards. The irrigation signals may present in the observations; however, the attribution of latent heat flux derived from the water balance model plays a more significant role in altering land surface temperature. The difference in albedo between the assumed irrigated and non-irrigated pixels results in a small temperature change. Throughout the growing season, this average difference in albedo is 0.00172, which has a weak effect on the land surface temperature (LST), contributing to a change of approximately 0.0116 K.

## 2.4 Identifying irrigated area

### 2.4.1 Classification method

Irrigated areas in the Rhine basin were identified with a combination of a hydrological model of wflow_sbm and satellite observations of land surface temperature as shown in Fig. 3. As illustrated in Fig. 3a, wflow_sbm does not physically represent irrigation practices. Meanwhile, satellite observations capture irrigation signals as an additional source of evapotranspiration in the water balance that modulates the partitioning of surface energy (Fig. 3b). This translates to higher latent heat flux and lower sensible heat flux than what the hydrological model predicts. Higher partitioning of available surface energy for latent heat flux results in lower land sur-

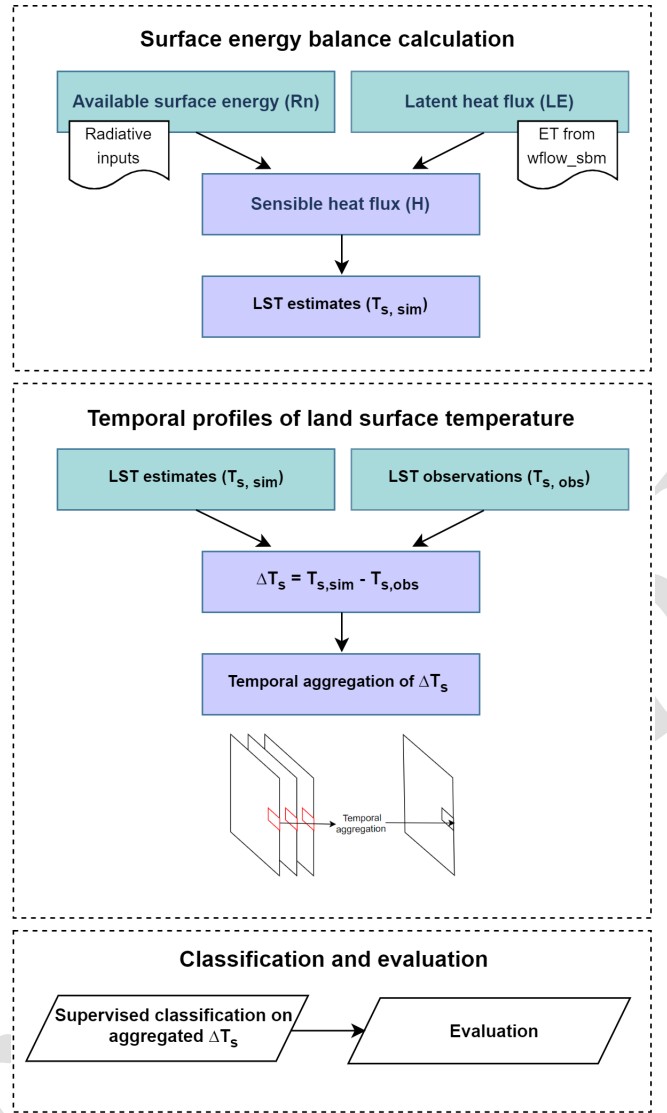

**Figure 2.** The workflow outlines the methodology for linking evapotranspiration estimates to derive land surface temperature. The spatiotemporal features of land surface temperature difference are used as input data to identify irrigated areas.

face temperature. Consequently, simulated land surface temperature data that were derived from the evapotranspiration estimate ($T_{s,sim}$) as described in Sect. 2.3 will be higher than the observed land surface temperature ($T_{s,obs}$). However, on a surface where there is no additional source of evapotranspiration, there are no changes in the energy and water balance fluxes. Figure 3d and e show the time series of $T_{s,sim}$, $T_{s,obs}$, and $\Delta T_s$, where an irrigated pixel in a hydrological model exhibits a higher magnitude of $\Delta T_s$ than a neighbouring non-irrigated cropland pixel, which remains relatively constant throughout the year. To classify irrigated area, the following data were used to compute $\Delta T_s$ and are listed below:

1. Land surface temperature at 1 km resolution as $T_{s,obs}$ from MODIS sensor on board Terra and Aqua (Wan et al., 2021a, b). MODIS observation data were resam-

pled and reprojected from sinusoidal to the geographic coordinate system.

2. Simulated actual evapotranspiration from wflow_sbm.

Cloud cover is prevalent in the daily LST observations of the study area. A statistical analysis was carried out to quantify the data gaps in the MODIS annual LST data cube during April to October from 2010 to 2019 caused by missing values from cloud cover. The results show a mean data gap due to cloud cover of approximately 59.3 % over the 10-year period for Terra and Aqua, with high seasonal variation. Cloud cover was highest during in April (67.7 %) and lowest during the peak of the growing season in July (48.4 %). Due to data gaps resulting from cloud cover and sampling frequency limitations in observations, yearly irrigation identi-

fication was made feasible by aggregating cloud-free daily $\Delta T_s$ over 1 year. Therefore, irrigated areas in this study are defined as pixels where irrigation is detected within a given year. In cases where irrigation events are recurrent within the same year, these events are counted as a single event.

To capture spatiotemporal features, we used statistical measures: $p_{10}$, $p_{50}$, $p_{90}$, mean, and standard deviation to aggregate equidistant observations into an annual data cube. The use of spatiotemporal features has been a common practice in previous irrigation mapping studies. For example, Dari et al. (2021) used the spatiotemporal dynamics of soil moisture, including day-to-day variability, as a feature in k-means clustering to distinguish between irrigated and non-irrigated land in the Mediterranean. After computing $\Delta T_s$, we applied the random forest algorithm by Breiman (2001) to classify irrigated and non-irrigated pixels. The results were screened to remove pixels that were identified as irrigated only once during the study period, as the installation cost of irrigation equipment is high. The resulting estimation distinguishes between irrigated and non-irrigated pixels and does not produce irrigation fraction of the entire pixel area.

### 2.4.2 Training and test dataset for random forest classification

Datasets for building random forest classification were acquired inclusively for each year for the period 2010–2019 to account for possible variations in the irrigated area due to climate conditions. Due to the unavailability of multiyear observation data for our purpose, we had to rely on true and thermal imagery with high spatial resolution to collect point data. To minimize errors in visual detection due to its subjectivity, we complemented the visual detection with thermal imagery that captures differences in land surface temperature signatures at the plot scale with similar meteorological conditions. Combining these methods can reduce the degree of uncertainty regarding the demarcation between irrigated and non-irrigated areas due to additional information provided by land surface temperature. Those datasets were collected from high-resolution imagery from Landsat 7 and 8 with a spatial resolution of 30 m (visible) and 100 m (thermal) as shown in Fig. 4. The datasets collected from this procedure are used as point labels for the classifier trained on $\Delta T_s$ data. The methodologies used in this step draw on heuristic techniques used in previous remote sensing studies (Peña-Arancibia et al., 2016; Deines et al., 2019; Shahriar Pervez et al., 2014), as elaborated below:

1. Point labels were collected using true-colour images captured during the growing season. These images were particularly valuable in identifying irrigated fields at the beginning of the growing season. During this specific period, visual identification of plots under irrigation or equipped with irrigation was feasible.

2. True-colour images were plotted concurrently with thermal observations to distinguish irrigated pixels from neighbouring pixels. Additionally, this prevents misinterpretation of pixels with darker soil resulting from ploughing as irrigated pixels. When such conditions are observed, these pixels are labelled as "non-irrigated". All training labels follow a binary classification that distinguishes between irrigated and non-irrigated pixels.

The time series of $\Delta T_s$ was also used to explore the potential presence of irrigated pixels. When potential irrigated pixels from Landsat true-colour and thermal images were identified, a noticeable increase in $\Delta T_s$ was observed. In cases where these temperature differences did not correspond to agricultural land parcels identified from the land cover map, it was inferred that these variations might arise from alternative sources or could be influenced by the presence of neighbouring land cover types, such as floodplain and forests. Consequently, the pixels were labelled as "non-irrigated". The dataset obtained from high-resolution imagery was divided into two subsets: 80 % for a training set and 20 % for a test set. The test set was used to assess the performance of the model, which was trained using the training data. Detailed information on the metrics used to evaluate the model and its performance is summarized in Appendix B.

### 2.4.3 Evaluation data

The implementation of a classification analysis using a random forest classifier has produced a series of 10 annual irrigation maps from 2010 to 2019. The validation of these maps involves both temporal and spatial assessments of the irrigated areas. Unfortunately, there are no datasets available for this purpose. Given the absence of ground-based observational data on irrigated areas, our multiyear classification assessment relies on comparisons with irrigation statistics. Specifically, national-level statistics regarding irrigated areas within the basin were obtained from the statistical office of the European Union, Eurostat, for the years 2013 and 2016 at the NUTS 2 (indicator: ef_poirrig). These statistics were sourced from the FSS, where differences in methodologies and variables between countries could cause potential uncertainties in the report. The data area is available at https://ec.europa.eu/eurostat/data/database (last access: 20 June 2024). Additionally, data on irrigated areas in Germany for 2019 were provided by the Federal Statistics Office of Germany. The classification results were evaluated for (i) overall, (ii) dry, and (iii) wet NUTS2 regions, which were defined based on the climatology of precipitation and potential evapotranspiration summarized in Table 1. The dry regions were classified as NUTS level 2 regions that lie within the Middle Rhine sub-basins. Meanwhile, the wet regions are in the Moselle, Neckar, Main, and Lower Rhine sub-basins. The comparison between the mapped area and re-

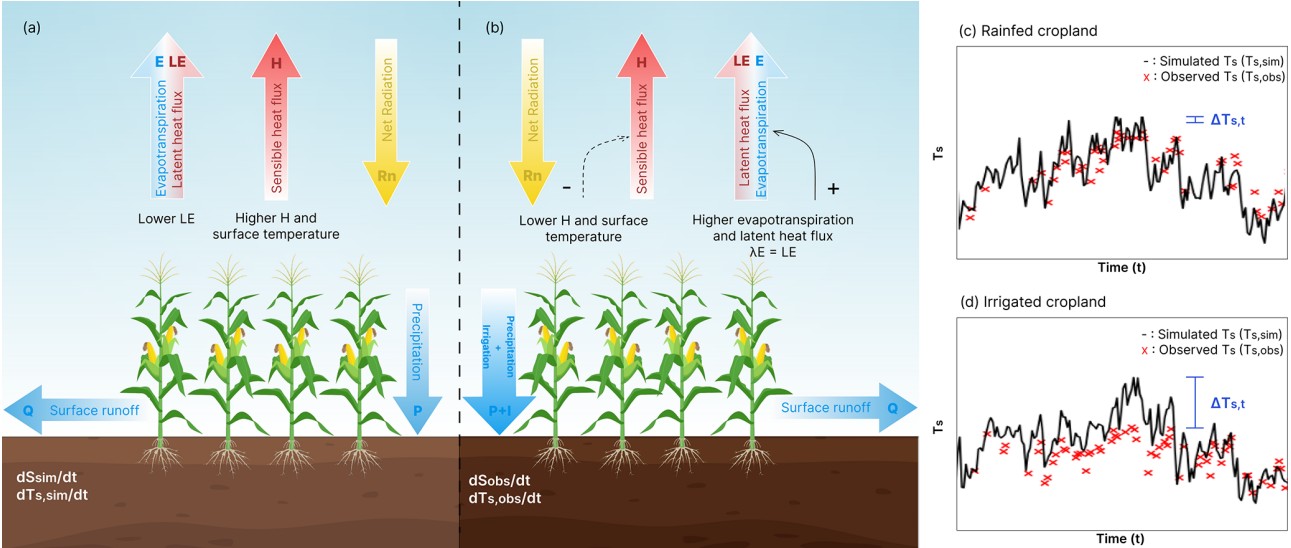

**Figure 3.** Schematic of the energy and water balance in **(a)** hydrological model wflow_sbm that does not represent irrigation practices and **(b)** earth observations that capture irrigation signals. In this study, we use land surface temperature observations. $T_{s,sim}$ refers to land surface temperature that is derived from sensible heat flux after relating evapotranspiration in water balance to latent heat flux in energy balance through $\lambda$. Irrigation increases the partitioning of available energy to latent heat flux, leading to lower $T_{s,sim}$. **(a)** The magnitude of $T_{s,obs}$ of the non-irrigated croplands is slightly similar to $T_{s,sim}$, where **(b)** $T_{s,obs}$ is lower than $T_{s,sim}$ due to higher evapotranspiration.

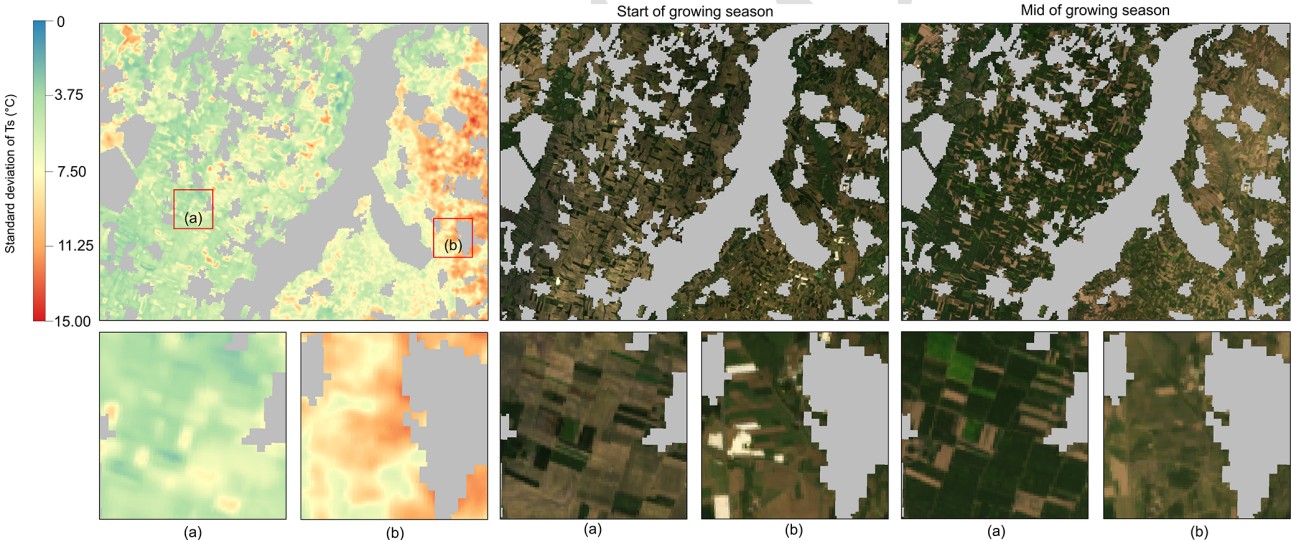

**Figure 4.** Illustration for training and test dataset: a snapshot of cropland area within the basin (the location is marked by a red rectangle in Fig. 1a) showing the seasonal standard deviation of land surface temperature ($T_s$) alongside true-colour images collected from Landsat annual cloud-free composite images (April–September). It reveals that irrigated areas in panel **(a)** exhibit a lower standard deviation in land surface temperature, whereas non-irrigated areas in panel **(b)** show a higher standard deviation. Shaded grey areas are masks for non-cropland land cover.

ported area for each NUTS 2 region was mapped to evaluate differences between datasets.

To further assess the consistency and accuracy of irrigated areas, the spatial distribution of the irrigated area was compared to the existing irrigated maps: Global Irrigated Area Map (GIAM) (Thenkabail et al., 2009), Global Map of Ir-

rigated Areas (GMIA) (Siebert et al., 2013), MIRCA2000 (Portmann et al., 2010), and Global Irrigated Area (Meier et al., 2018), as summarized in Table 2. The first three products were developed at a 5 arcmin resolution, while the latter was developed at a 1 km spatial resolution. MIRCA2000 and GMIA used sub-national statistics and geographical in-

formation on the location of irrigation schemes as references to produce maps detailing irrigation portion. Meanwhile, GIAM and Global Irrigated Area made use of remote sensing products and techniques to provide irrigation maps in binary format. Specifically for the Global Irrigated Area, it used NDVI to downscale the distribution of irrigation indicated in GMIA.

# 3 Results

## 3.1 Land surface temperature from hydrological modelling

Figure 5 shows an example temporal profile of the average basin precipitation, evapotranspiration, and land surface temperature for both irrigated (Fig. 5b) and non-irrigated pixels (Fig. 5c) for training data. Despite high precipitation from January to the end of April, the low potential evapotranspiration during this period does not contribute to an additional source of latent heat flux due to limited available surface energy. As a result, $T_{\text{s,sim}}$ for both irrigated and non-irrigated pixels closely resembles $T_{\text{s,obs}}$. However, differences between $T_{\text{s,sim}}$ and $T_{\text{s,obs}}$ become more apparent in irrigated pixels as potential evapotranspiration gradually increases from the beginning to the peak of the growing season, reaching differences of up to approximately $10\,°\text{C}$. Following the peak, $\Delta T_{\text{s}}$ gradually declines towards the end of the growing season corresponding to the potential evapotranspiration rate with a lag. In contrast, $\Delta T_{\text{s}}$ of non-irrigated pixels remains relatively constant during the growing season despite the gradual increase in potential evapotranspiration. As $\Delta T_{\text{s}}$ gradually increases towards the peak of growing seasons on irrigated pixels, it leads to higher annual $\Delta T_{\text{s}}$ variability compared to non-irrigated pixels. This observed $\Delta T_{\text{s}}$ across irrigated pixels suggests the presence of other sources of evapotranspiration which were not considered in the model.

These distinct daily temporal patterns of $\Delta T_{\text{s}}$ between irrigated and non-irrigated pixels were used to estimate annual irrigation extent. Figure 6 shows an example of statistical summaries of $\Delta T_{\text{s}}$ for irrigated and non-irrigated pixels in 2018 and 2019. Small fractions of data points with negative $\Delta T_{\text{s}}$ due to random error in Fig. 5 are represented by $p_{10}$. This has minimal influence on the classification results due to the similar magnitude of $\Delta T_{\text{s}}$ between irrigated and non-irrigated pixels. Except for $p_{10}$, irrigated pixels show higher $p_{50}$, $p_{90}$, mean, and standard deviation relative to non-irrigated land due to different temporal profiles of $\Delta T_{\text{s}}$. These differences in statistical summaries between irrigated and non-irrigated pixels are more pronounced in dry years than in wet years, resulting in varying magnitudes in statistical summaries throughout different years. From this information, a model trained with data from a specific year cannot be used to identify irrigated areas for the whole study period due to varying meteorological conditions.

## 3.2 Interannual variability in irrigated area

Figure 7 shows the comparison between the reported irrigated areas from Eurostat data and the mapped irrigated areas for the years 2013 (Fig. 7a) and 2016 (Fig. 7b). As the linear fit is strongly influenced by regions with large irrigated areas, the datasets were transformed using a logarithmic transformation to assess the difference between the estimated and reported values in regions with limited irrigated areas. Overall, the mapped irrigated areas at NUTS level 2 show a good agreement with the reported irrigated areas, with $R_{\text{oa}}^2$ values of 0.79 and 0.77 for 2013 and 2016, respectively. The mapping methodology performs slightly better in dry regions than in wet regions. For dry regions, the $R_{\text{dr}}^2$ values are 0.9 and 0.87, while, for wet regions, the $R_{\text{wr}}^2$ values are 0.705 and 0.783 for 2013 and 2016, respectively, an average improvement of 0.14. In some NUTS level 2 regions for both years, the mapped irrigated areas exceed the reported irrigated area, with an average percentage relative difference of 17 % (ranging from 12 % to 22 %). The overestimation of the irrigated area is more prevalent in wet regions for both years. The seemingly large underestimation of Upper Franconia (DE24) in 2013 and overestimation of Kassel (DE73) in 2016 are influenced by the logarithmic scale, which exaggerates the reported and predicted values. The underestimation is $\sim 34\,\text{ha}$, and the overestimation is $\sim 54\,\text{ha}$, both of which fall below the detection threshold of spatial resolution. The overestimation of irrigated area is particularly notable in regions characterized by small-scale irrigation holdings where irrigation is sparsely distributed alongside mixed land use, such as Koblenz (DEB1), Middle Franconia (DE25), Tübingen (DE14), and Arnsberg (DEA5). Based on statistics reported by the Federal Statistical Office of Germany, these regions have an average irrigated area of 5–9 ha per agricultural holding in 2019. The mapping methodology performed better in regions characterized by large irrigation holdings (with an average $> 22\,\text{ha}$ per holding), such as Alsace (FR42), Rheinhessen-Pfalz (DEB3), Düsseldorf (DEA1), Darmstadt (DE71), and Cologne (DEA2).

The same mapping methodology was applied to identify irrigated areas, providing details on the extent of irrigation in the Rhine basin from 2010 to 2019. Based on the average from 10 annual maps, the irrigated area in the Rhine basin was estimated to be 159 000 ha, with the spatial distribution covering an area of 370 000 ha, as shown in Fig. 8. The irrigated areas were concentrated near Düsseldorf (DEA1), Cologne (DEA2), and Münster (DEA3) in the Lower Rhine region (Fig. 8b); Darmstadt (DE71) and Rheinhessen-Pfalz (DEB3) in the Main region (Fig. 8c); and Alsace (FR42) in the Middle Rhine region (Fig. 8d). Analysis of multiyear irrigated maps revealed that approximately 10 000 ha was consistently identified as receiving irrigation and was mostly

**Table 2.** Existing irrigation datasets to evaluate estimated irrigation extent of the Rhine basin.

| Products | Resolution | Period | Coverage | Methods | Source |
|---|---|---|---|---|---|
| Global Irrigated Area Map (GIAM) | 5 arcmin | A single map, 2000 | Global | Spectral matching techniques of remote sensing products | Thenkabail et al. (2009) |
| Global Map of Irrigated Areas (GMIA) v5.0 | 5 arcmin | Single map, representative for the period 2000–2008 | Global | Sub-national agricultural statistics and geographical information | Siebert et al. (2013) |
| MIRCA2000 | 5 arcmin | Single map, representative for the period 1998–2002 | Global | Sub-national agricultural statistics, harvested area, GMIA, and ancillary data | Portmann et al. (2010) |
| Global Irrigated Area | 1 km | Single map, representative for the period 1999–2012 | Global | Decision tree, NDVI, agricultural suitability, GMIA | Meier et al. (2018) |
| Eurostat statistics | Regional statistics of area irrigated at least once a year at NUTS 2 | 3-year interval (2013, 2016) | European Union | Farm Structure Survey (FSS) | https://ec.europa.eu/ eurostat/data/database |

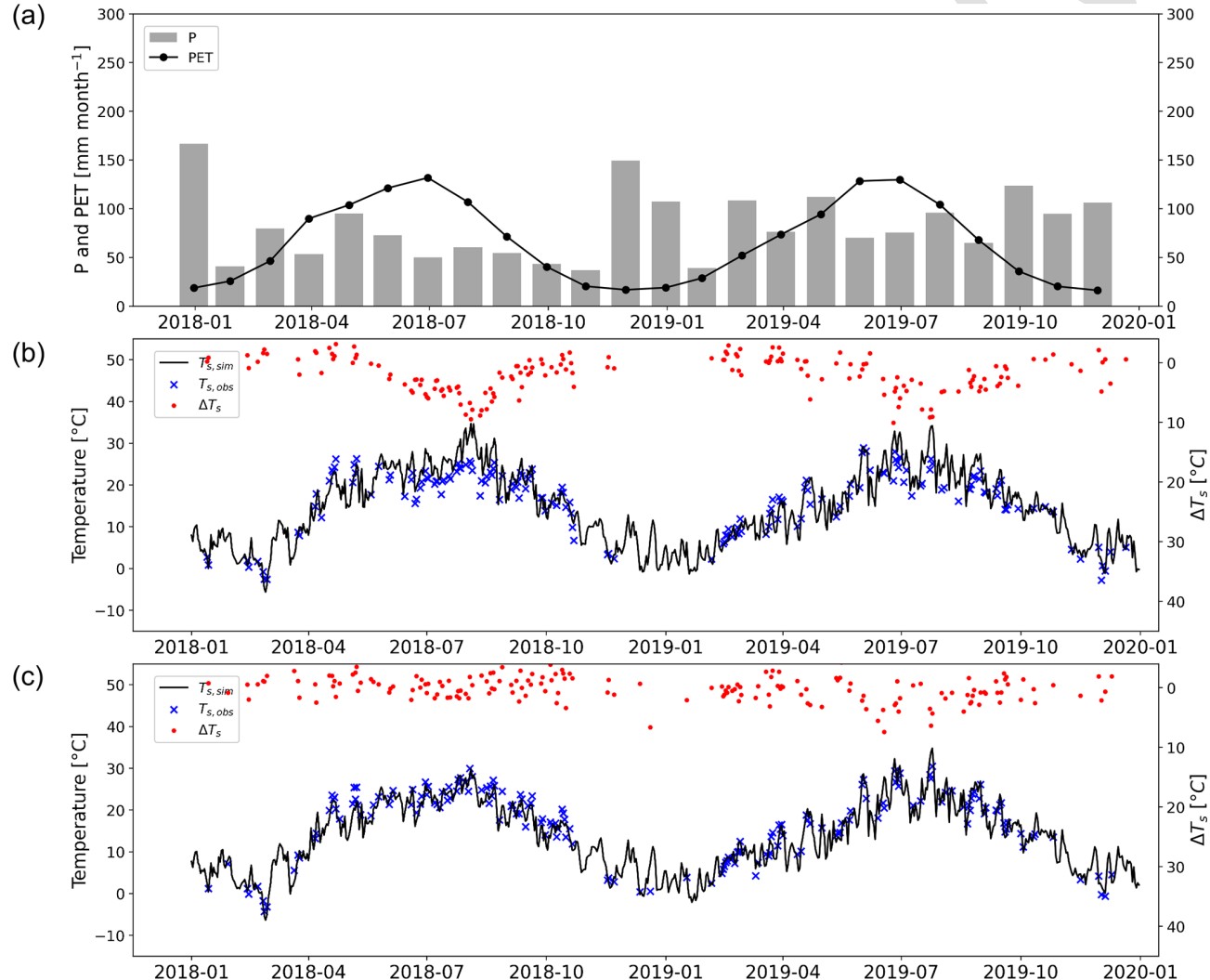

**Figure 5. (a)** Time series of monthly precipitation and potential evapotranspiration averaged across the basin alongside time series of simulated land surface temperature ($T_{s,sim}$), observed land surface temperature from MODIS ($T_{s,obs}$), and temperature difference $\Delta T_s$. These are provided for pixels regarded as **(b)** irrigated and **(c)** non-irrigated.

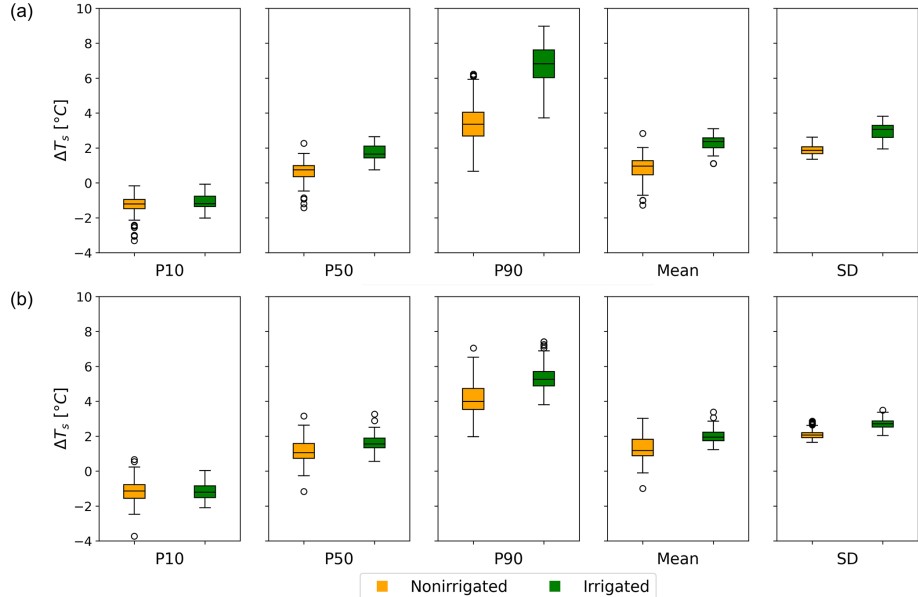

**Figure 6.** The box plot shows a statistical summary of training data for non-irrigated and irrigated pixels for **(a)** 2018 and **(b)** 2019.

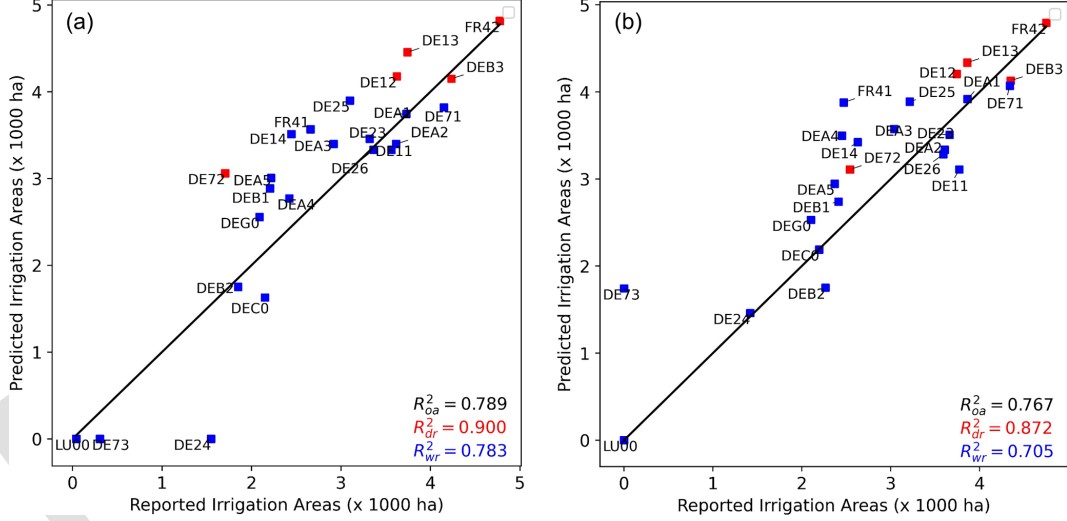

**Figure 7.** The mapped irrigated area of the Rhine basin as identified through classification ($A_{i,sim}$) is compared with the total irrigated areas reported in Eurostat data at NUTS level 2 ($A_{i,obs}$) for the years **(a)** 2013 and **(b)** 2016. $R^2$ values were calculated for the overall regions ($R^2_{oa}$), dry regions ($R^2_{dr}$), and wet regions ($R^2_{wr}$). The values of the total irrigated areas [$\times$ 1000 ha] were transformed using $\log(A+1)$ transformation.

found in Alsace. The mapped irrigated area at 1 km resolution allows the observation of additional information that is difficult to identify in irrigated products with coarser spatial resolution. For instance, in the Rhine valley, the spatial distribution of irrigated areas is predominantly concentrated to the west of the French–German border in the Alsace region, with higher density compared to neighbouring agricultural lands in Freiburg.

The spatial and temporal distribution of irrigated areas is influenced by irrigation management practices, which are

partially driven by climatic factors such as precipitation and evapotranspiration. At the basin level, there is a positive correlation between annual irrigated area and precipitation. However, Fig. 7 highlights challenges in irrigation identification in more humid regions. As classification performance in dry regions is higher than in more humid conditions, we use the Alsace region as an example of how climatic factor has an influence on irrigated areas, as it has the highest irrigated area in the region with an average of 65 860 ha. Figure 9 shows the correlations between precipitation and evapotran-

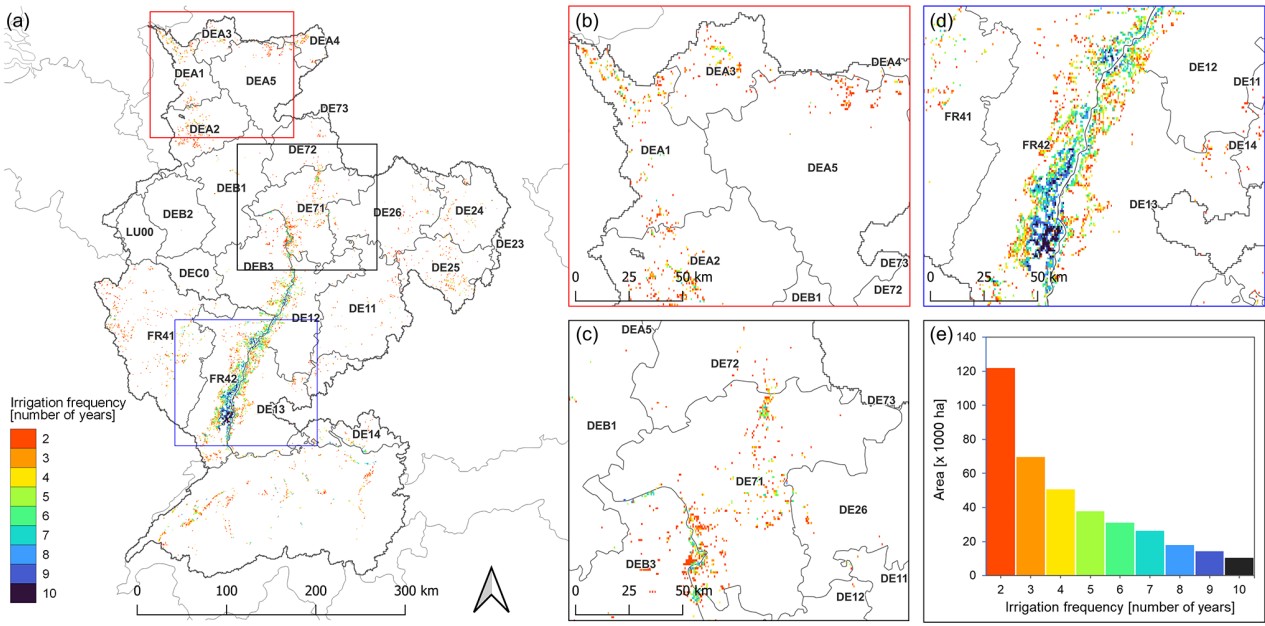

**Figure 8.** The extent of irrigated area derived from land surface temperature difference and irrigation frequency from the period 2010–2019. The rectangles in panel **(a)** show irrigation hotspots in **(b)** the Lower Rhine, **(c)** the Middle Rhine, and **(d)** the Rhine valley. Panel **(e)** shows the irrigation frequency and corresponding area.

spiration and their difference in yearly total irrigated areas. The analysis reveals an increase in total irrigated area during years with low precipitation ($r = -0.82$, $p$ value = 0.004). In 2011, 2015, and 2018, the Rhine basin experienced lower annual precipitation coupled with higher evapotranspiration compared to the previous year.

### 3.3 Intercomparison with existing irrigated maps

The identified irrigated areas are mainly found in the already known irrigation scheme in the current maps with additional identified irrigated areas as shown in Fig. 11. Potential discrepancies between existing products used in this study would be expected because of underlying differences in spatial resolution, input data, reference period, and processing techniques to derive irrigated areas. Our estimated irrigated area, which averages 159 000 ha, exceeds the actual irrigated area (AEI) reported by GMIA (148 000 ha) (Meier et al., 2018) and MIRCA2000 (110 000 ha) (Portmann et al., 2010). MIRCA2000 not only provides lower estimates for irrigated areas compared to GMIA, but also fails to accurately identify irrigated areas within the Main basin and some part of the Lower Rhine basin which were also reported in sub-national statistics from Eurostat. Although both use subnational statistics as a reference, MIRCA2000 determines irrigated areas based on maximum monthly irrigated area that was estimated based on crop-specific harvested area from Monfreda et al. (2008) as input data. Thus, a significant harvested area that was not reported in the crop-specific har-

vested area data may not have been properly distributed as an irrigated area (Portmann et al., 2010).

Our estimates are slightly lower than those provided by the Global Irrigated Map, which identified 21 000 ha of irrigated area using remote sensing products. The Global Irrigated Map distributed the irrigated area based on previous knowledge from the GMIA dataset. It was anticipated that the estimates from the Global Irrigated Map would be higher, given its use of higher spatial resolution and recent satellite observations to capture finer details. This resolution allowed us to identify denser irrigation in regions already identified as irrigated in the GMIA dataset and to discover newly irrigated croplands in regions previously not identified as irrigated (Meier et al., 2018). While, in some NUTS 2 regions, both our estimates and the Global Irrigated Areas dataset show higher irrigated areas compared to other existing maps, the locations of these irrigated pixels vary between the two maps (Fig. 12). Additional irrigated areas were identified in Freiburg, which is located to the east of the French–German border. This could be because irrigation is only used as a supplementary measure on crops during dry periods. Therefore, it is possible that the irrigated data from the Global Irrigated Area, which represents irrigation from 1999 to 2012, do not accurately represent irrigation dynamics during the study period.

In contrast, our estimates of irrigated areas are lower compared to those provided by the Global Irrigated Area Map (GIAM), which estimates an exceptionally high value of around $1.4 \times 10^6$ ha. The high value of GIAM estimates can

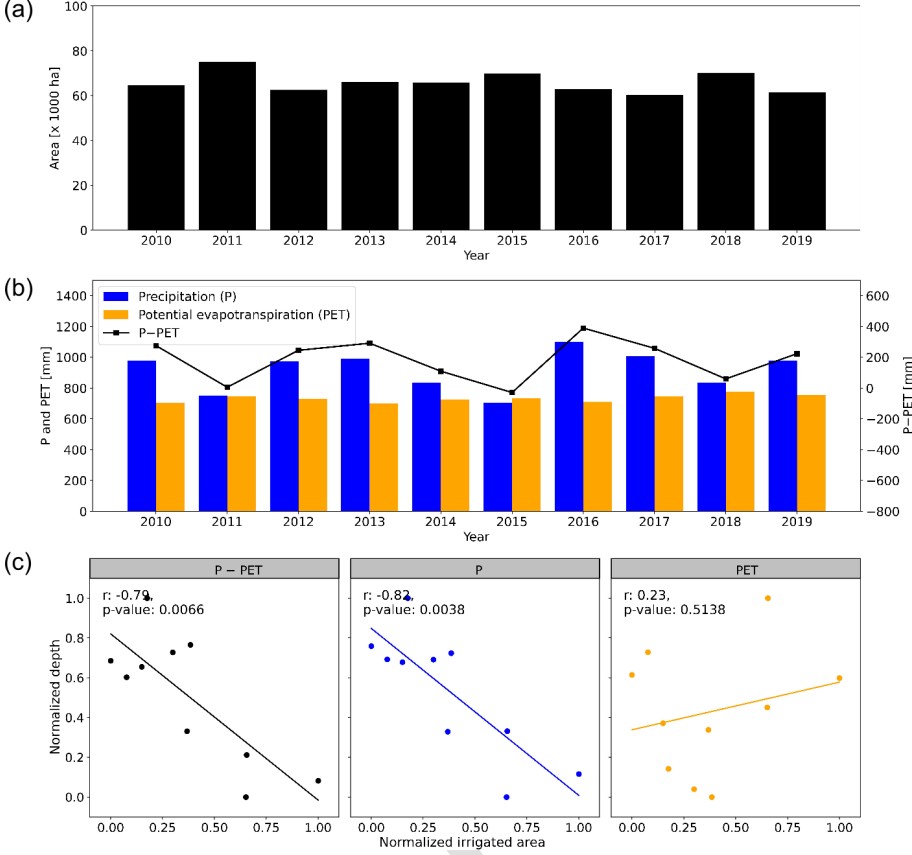

**Figure 9. (a)** The total irrigated area and **(b)** the annual sum of climatic variables: precipitation, evapotranspiration, and the difference for Alsace in the period from 2010 to 2019. **(c)** Linear regression analysis is performed for each climatic variable compared to the annual irrigated area.

be attributed to overestimation in the eastern part of the basin. However, it underestimates the irrigated areas in the Rhine valley, which is identified as the most heavily irrigated area in the basin in other products. This serves as an example that a different approach, different spatial resolutions, and different input data to identify the irrigated map can yield different results. Additionally, the reference period of existing products varies, which may not be representative of the period used in this study. The difference in the identified irrigated area was also experienced by Meier et al. (2018), who used GMIA data from 2005 to identify the irrigated area that is representative for the period 1999–2012.

## 4 Discussion

The results of this study demonstrate the potential of using evapotranspiration estimates from a spatially distributed hydrological model and satellite observations of land surface temperature to detect and monitor irrigated areas. Irrigation modulates the partitioning of surface energy and water balance through evapotranspiration which leads to reductions in land surface temperature in irrigated croplands. These im-

pacts of irrigation on land surface temperature were also used in previous regional studies to identify irrigated areas (Shahriar Pervez et al., 2014; van Dijk et al., 2018). By coupling surface energy with water balance in the model, we can improve the identification of irrigated areas, particularly in regions where precipitation patterns coincide with irrigation cycles. Although our estimates were produced without relying on existing maps to determine the location of irrigated areas, the proposed methodology can reasonably approximate the extent of irrigated areas when evaluated against existing irrigation maps (Fig. 7).

Mapping irrigated plots at the catchment level in our study region presents challenges due to insufficient distinct features between irrigated and non-irrigated areas during dry years and even more so in years with adequate precipitation, when non-irrigated croplands exhibit the same LST temporal features as irrigated croplands (Appendix C). By using $\Delta T_s$ obtained from observations and hydrological models, evapotranspiration from precipitation estimated through water balance can be excluded, isolating only the evapotranspiration driven by irrigation. In our study area, the temporal patterns of $\Delta T_s$ provide more distinctive features for clas-

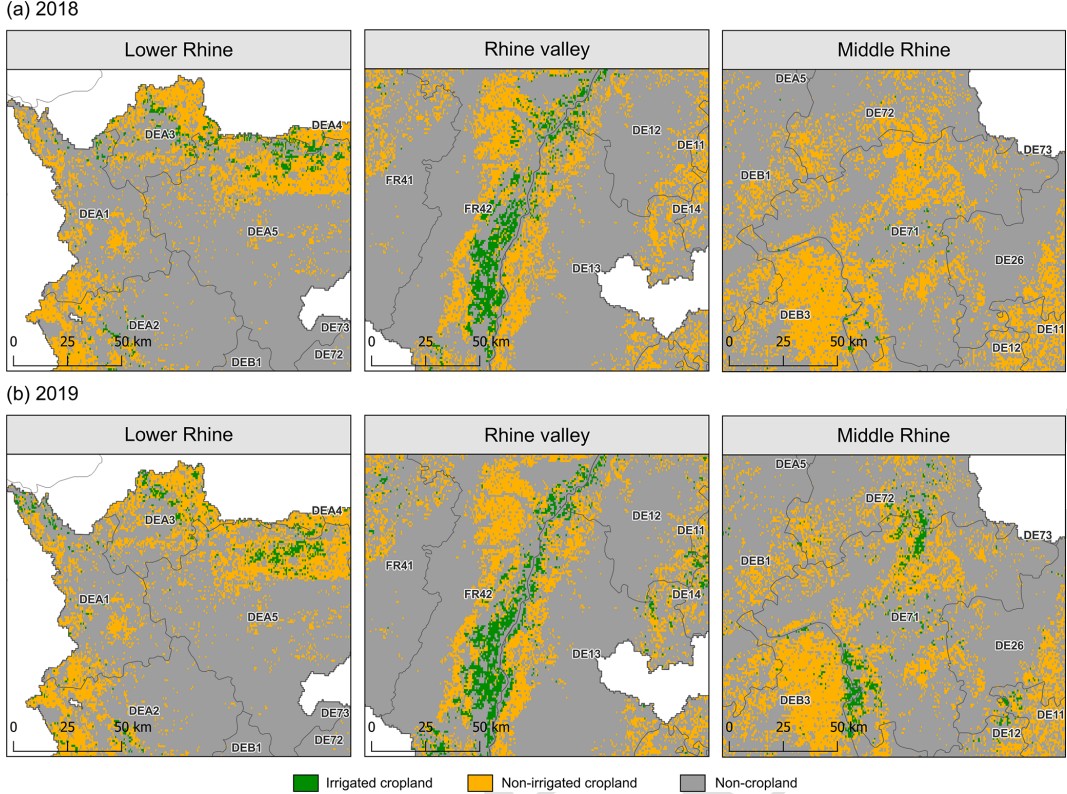

**Figure 10.** Difference in the extent of irrigated area between **(a)** 2018 and **(b)** 2019 for the Lower Rhine, the Rhine valley, and the Middle Rhine. TS1

sification compared to using LST alone. However, the proposed methods still face challenges related to the interannual variability in $\Delta T_s$, which results in a year-specific model. Several reasons may be due to the dynamic nature of irri-
5 gation decisions, fallow practices, and the interannual variability in meteorological conditions. A model trained on data from a single year may fail to account for these variabilities, as it uses LST features or thresholds from irrigated or non-irrigated pixels in years with differing conditions.

10 The difference between our estimates and the irrigated area reported by official statistics can be attributed to two main factors: (i) the spatial resolution difference and (ii) uncertainties in the reported irrigated areas. In our classification process, we do not adjust the area of a pixel identified as
15 either irrigated or non-irrigated based on the size of agricultural holdings in the region, which may lead to bias in regions where agricultural holdings smaller than $1\,\mathrm{km}^2$ are dominant. Meanwhile, the reported irrigated areas from Eurostat were collected through questionnaires distributed to several agri-
20 cultural holdings. Comparing continuous spatial information from classification results with point information obtained from questionnaires is not ideal. The scaling issues between these two types of data make direct comparison difficult and can lead to misinterpretation of the extent of irrigation in
25 the region. Such disparities between the spatial resolution of

mapping units and the actual size of irrigation plots in the field may lead to the identification of additional areas as irrigated lands (Colombo et al., 2008). Additionally, validating our maps poses challenges because of potential errors in the data collected from the FSS of 2013 and 2016. These sur- 30 veys are subject to both sampling and non-sampling errors. The FSS data collection involves random sampling methods and extrapolation techniques, potentially resulting in deviations between the randomized sampling result and the true value of the entire population (Eurostat, 2016). 35

To resolve fragmented irrigated areas, finer-resolution maps are usually used, as they offer fewer mixed signals over regions with heterogeneous land cover types (Velpuri et al., 2009). However, this comes with a trade-off in terms of longer data processing times. Although a spatial resolu- 40 tion of $1\,\mathrm{km}^2$ is suitable for water management at the basin level and performed well in the study area, it may not be able to capture irrigated areas in regions with significant small fragmented agricultural holdings and heterogeneous land use (Fig. 7). This underscores the necessity of including method- 45 ology for irrigated area estimations in regions characterized by fragmented agricultural holdings (i.e. sub-pixel calculations in the Global Irrigated Area Map (GIAM) (Thenkabail et al., 2009) or regional field size factor (Salmon et al., 2015)). Nevertheless, the approach to determine these fac- 50

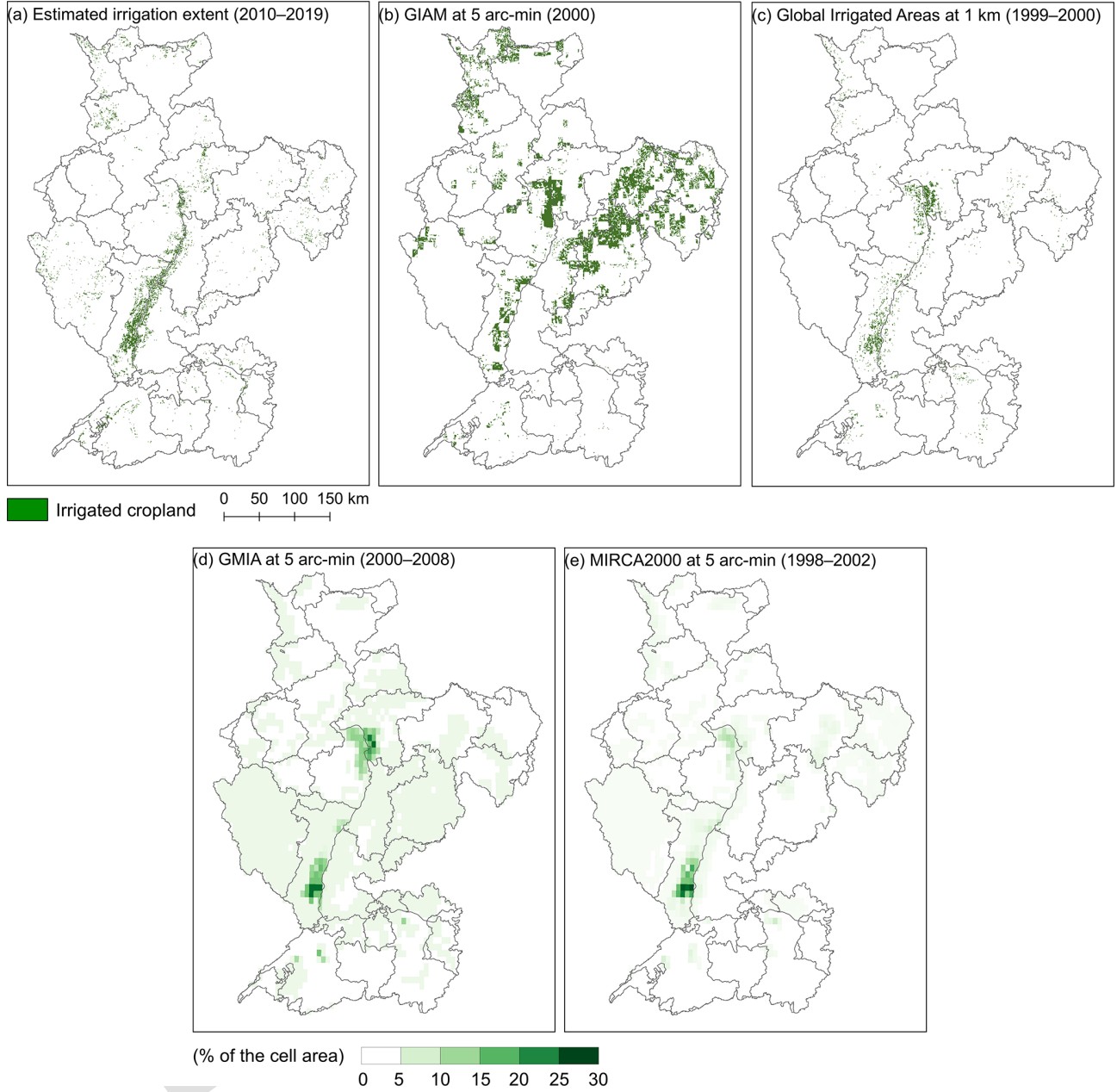

**Figure 11.** Comparison the estimated irrigation extent using land surface temperature with current irrigation maps.

tors requires validation, as it may introduce uncertainties in the outcome (Meier et al., 2018).

Additional uncertainties are also attributed to the input datasets and methodology. The input datasets of our study consist of evapotranspiration estimates from a hydrological model and satellite observations of land surface temperature. Since satellite observations implicitly capture various types of evapotranspiration, the parameterization (i.e. soil parameters, rooting zone) within the hydrological model to estimate evapotranspiration could yield land surface temperature estimates that do not accurately indicate irrigation.

A study by van Dijk et al. (2015) demonstrates that satellite observations captured additional evapotranspiration from groundwater-dependent ecosystems, which is not attributed to precipitation. This justifies the decision to mask out wetlands and forests to eliminate additional sources of evapotranspiration, such as lateral inflow and deep root water intake, before applying the algorithm, as these processes can produce a misleading indication of irrigation. Misclassification in CORINE land cover and land-use data, which were used to mask out non-cropland pixels for the classification process, introduces further uncertainties. Despite the high ac-

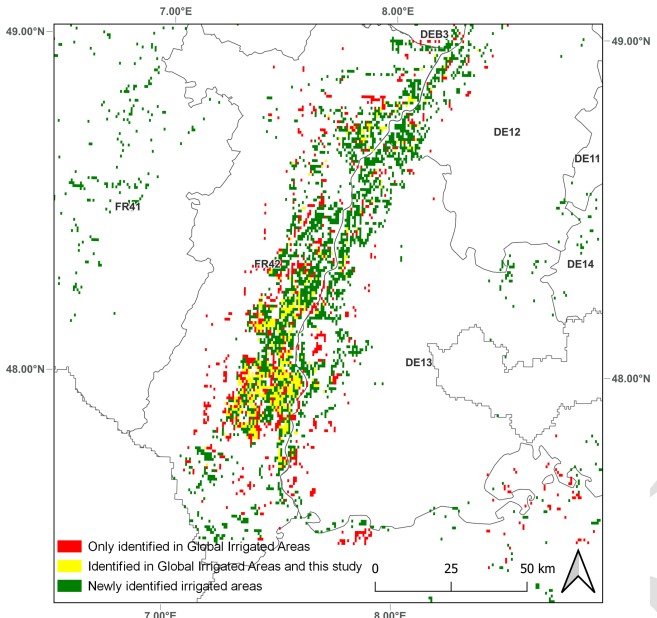

**Figure 12.** Irrigated areas in the Rhine valley. The green areas are the newly identified irrigated areas, the yellow areas are irrigated areas identified both in this study and in the Global Irrigated Area by Meier et al. (2018), and the red areas are irrigated areas which were only identified in the Global Irrigated Area.

curacy of the land-use data, occurrences of false classification were observed (not shown), thereby propagating error to our estimates of irrigated areas. In particular, mixed-land-use areas where pasture and cropland are difficult to map are likely to have higher error rates due to misclassification. Furthermore, the absence of pixel area fractions in the cropland data sourced from the land-use land cover dataset may potentially lead to an overestimation of the irrigation area.

While the proposed method performs reasonably well at the basin level, challenges remain in accurately detecting irrigated areas in humid regions, as highlighted by Zhang et al. (2022). Lower performance during wet regions can be partially explained by temporal dynamics of $\Delta T_s$ that showed less variability in wet years than in dry years (Roth et al., 2013). In the Alsace region, where irrigation is prevalent, decreased precipitation leads to an increase in the extent of irrigated areas during the driest years. This observed trend contrasts with most studies conducted in arid to semi-arid regions (e.g. Afghanistan (Shahriar Pervez et al., 2014) and the High Plains Aquifer (Deines et al., 2019)), which highlight the impact of limited water availability on irrigation decision-making. Decreased irrigated areas in arid to semi-arid regions can be explained by Foster et al. (2014), who demonstrate that farmers often prioritize maintaining soil water availability to minimize the risk of significant production losses by concentrating water supply on a smaller area. This irrigation strategy is constrained by regulatory restrictions that limit water abstraction. Therefore, our finding suggests that farmers increasingly rely on irrigation during periods of reduced precipitation to mitigate the risk of yield loss.

This highlights the need to further evaluate how much pressure from irrigation water use there is on water availability during drought. Although other factors influencing irrigation dynamics, such as improvements in irrigation efficiency, regulations, and restrictions on groundwater, were not studied, they may significantly influence the temporal dynamics of irrigation and need to be investigated.

Although the total irrigated area comprises only about 2 % of the total basin, peaks in land surface temperature differences were observed during the summer months (JJA), when precipitation cannot compensate high crop evapotranspiration. This translates to high irrigation rates being applied to offset the high rate of crop evapotranspiration, which puts additional pressure on limited water availability. Under changing climate conditions, projections for the Rhine basin indicate that a combination of changes in snow melting processes and increased potential evapotranspiration will result in decreased summer discharge (Buitink et al., 2021). This scenario highlights the urgency of addressing irrigation water demands and potential water deficits during summer months. However, these areal expansions and/or reductions throughout the study period were only detected in agricultural land cover, since the classification was performed within the agricultural class. Thus, any changes in land use and land cover were not accounted for in the results. It should be mentioned that the evaluation was performed over a simulation period of 10 years ($N = 10$) and that a longer time series will likely reduce random error (Thiese et al., 2016).

## 5 Conclusions

We used an energy balance approach to identify irrigated areas using land surface temperature derived from the evapotranspiration of a hydrological model and land surface temperature products from MODIS. The proposed methodology was able to identify irrigated areas in the Rhine basin, showing good agreement with sub-national statistics. However, the performance of the model deteriorates when applied to regions with small fragmented agricultural areas due to differences between the spatial resolution of mapping units and the actual size of irrigation plots (Salmon et al., 2015; Shahriar Pervez et al., 2014). When evaluated against existing irrigation maps, our results show underestimation and overestimation, which can be attributed to spatial resolution, input data, reference period, and processing techniques. Although technically feasible, comparing our estimate of irrigated area with other irrigation maps would not necessarily mean validation, as those maps have typically not undergone comprehensive validation against actual ground observations.

The results of our study reveal annual variability in irrigated areas, highlighting the necessity of gathering multi-year data to improve water resource management. In regions where irrigation is dominant, these variations in irrigation area are attributed to precipitation, with the irrigated area increasing during dry years. While our study does not evaluate other contributing factors besides climatic variables, such as policy measures, previous studies demonstrate the influence of regulatory frameworks on irrigation water use, which need to be studied. Challenges in irrigation detection in humid areas where the classification method performs slightly worse than in dry regions. This can be explained by less variability in LST in this region.

Uncertainties and limitations are inherent in our results. Uncertainties could be introduced through the classification process, input data, spatial resolution, and evapotranspiration products from the hydrological model. It should be noted that our approach currently predicts annual irrigated areas due to limitations imposed by the availability of thermal imagery. This constraint complicates the applicability of our method for weekly or even daily observations. Thus, considering the temporal resolution of land surface temperature data becomes important, as enhancing this resolution has the potential to improve the methodology for identifying irrigated areas, particularly in regions where precipitation occasionally aligns with irrigation cycles.

## Appendix A:  Land surface temperature module

The latent heat vaporization $\lambda$ equals

$$\lambda = 2501 - 2.375 T_\mathrm{a}. \tag{A1}$$

The net radiation $R_\mathrm{n}$ is calculated from radiation components from satellite observations, which are calculated as

$$R_\mathrm{n} = R_\mathrm{s}^\mathrm{in} - R_\mathrm{s}^\mathrm{out} + R_\mathrm{l}^\mathrm{in} + R_\mathrm{s}^\mathrm{in}, \tag{A2}$$

$$R_\mathrm{n} = R_\mathrm{s}^\mathrm{n} + R_\mathrm{l}^\mathrm{n}. \tag{A3}$$

The amount of outgoing shortwave radiation $R_\mathrm{s}^\mathrm{out}$ that is reflected to space is determined by the surface albedo $\alpha$. Therefore, to account for the energy loss from the outgoing shortwave radiation, the net shortwave radiation $R_\mathrm{s}^\mathrm{n}$ is calculated using the following formula:

$$R_\mathrm{s}^\mathrm{n} = (1 - \alpha) R_\mathrm{s}^\mathrm{in}. \tag{A4}$$

The rate of energy loss from the outgoing longwave radiation $R_\mathrm{l}^\mathrm{out}$ is determined by the Stefan–Boltzmann law, where the Stefan–Boltzmann constant $\sigma = 5.67 \times 10^{-8} \, \mathrm{W\,m^{-2}\,K^{-4}}$. The estimates of net longwave radiation are then calculated by adjusting the outgoing longwave radiation based on humidity and cloudiness, as these factors impact the absorption and reflection of radiation fluxes (Allen et al., 1998).

$$R_\mathrm{l}^\mathrm{n} = (\sigma T_\mathrm{a}^4)(0.34 - 0.14\sqrt{ea}) \left(1.35 \frac{R_\mathrm{s}^\mathrm{in}}{R_\mathrm{so}} - 0.35\right), \tag{A5}$$

$$e_\mathrm{a} = 0.611 \exp\left(\frac{17.27 T_\mathrm{a}}{(T_\mathrm{a} + 273.3)^2}\right) \tag{A6}$$

The expression $(0.34 - 0.14\sqrt{ea})$ represents the impact of humidity on the net outgoing longwave radiation. The term $1.35 \frac{R_\mathrm{s}^\mathrm{in}}{R_\mathrm{so}}$ expresses the impact of cloudiness on incoming shortwave radiation, where $R_\mathrm{so}$ can be calculated as follows:

$$R_\mathrm{so} = 0.75 R_\mathrm{a}, \tag{A7}$$

$$R_\mathrm{a} = G_\mathrm{sc} d_\mathrm{r}(\omega_\mathrm{s} \sin\phi \sin\delta + \cos\phi \cos\delta \sin\omega_\mathrm{s}), \tag{A8}$$

where the magnitude of extraterrestrial radiation $R_\mathrm{a}$ is determined based on solar constant, the inverse relative distance from the Earth to the Sun is $d_\mathrm{r}$, the sunset hour angle is $\omega_\mathrm{s}$, the latitude is $\phi$, and the solar declination is $\delta$. Therefore, the sensible heat flux equation becomes

$$H = (1 - \alpha) R_\mathrm{s}^\mathrm{in} + (\sigma T_\mathrm{a}^4)(0.34 - 0.14\sqrt{ea})$$
$$\times \left(1.35 \frac{R_\mathrm{s}^\mathrm{in}}{R_\mathrm{so}} - 0.35\right) - LE. \tag{A9}$$

The aerodynamic resistance $r_\mathrm{a}$ that governs the vapour and heat transfer is computed based on Thom's equation (Thom, 1975) and roughness parameters recommended by Allen et al. (1998):

$$r_\mathrm{a} = \frac{\ln(\frac{z_\mathrm{m} - d}{z_\mathrm{om}}) \ln(\frac{z_\mathrm{h} - d}{z_\mathrm{oh}})}{k^2 u_z}, \tag{A10}$$

$$d = \frac{2}{3} h_\mathrm{c}, \tag{A11}$$

$$z_\mathrm{om} = 0.123 h_\mathrm{c}, , \tag{A12}$$

$$z_\mathrm{oh} = 0.1 z_\mathrm{om} \tag{A13}$$

where $d$ is the zero plane displacement height, $z_m$ is the height of wind measurement, $z_h$ is the height of humidity measurement, $z_{oh}$ is the roughness length of vapour and heat transfer, $z_{om}$ is the roughness length of momentum transfer, $u_z$ is the wind speed measured at the height of 2 m, $h_c$ is the crop height, and the von Kármán constant $k = 0.41$. In this study, the heights of measurements for wind and humidity are assumed to be equal ($z = z_m = z_h$). During periods of extremely low wind conditions, the wind speed is constrained to be greater than $0.5\,\mathrm{m\,s^{-1}}$ to consider vapour exchange on the surface induced by air buoyancy and layer instability effects (Allen et al., 1998).

## Appendix B: Random forest performance on test data

The performance of the random forest model was evaluated using several performance evaluation metrics which are obtained from true negatives (TNs), true positives (TPs), false negatives (FNs), and false positives (FPs). The recall measures the portion of irrigated areas in a test set which were correctly identified. The precision shows the portion of pixels identified as irrigated which are actually irrigated. The $F_1$ score combines both recall and precision into a unified metric:

$$\text{accuracy} = \frac{\Sigma(\text{TP} + \text{TN})}{\Sigma(\text{TP} + \text{FP} + \text{TN} + \text{FN})}, \tag{B1}$$

$$\text{recall} = \frac{\Sigma \text{TP}}{\Sigma(\text{TP} + \text{FN})}, \tag{B2}$$

$$\text{precision} = \frac{\Sigma \text{TP}}{\Sigma(\text{TP} + \text{FP})}, \tag{B3}$$

$$F_1 = \frac{2 \times \text{recall} \times \text{precision}}{\text{recall} + \text{precision}}. \tag{B4}$$

**Table B1.** The accuracy, precision, recall, and $F_1$ score of the random forest model on training data used to classify irrigated areas based on land surface temperature differences for the years 2010 to 2019.

| Year | Accuracy | Precision | Recall | $F_1$ |
|------|----------|-----------|--------|-------|
| 2010 | 0.941 | 0.934 | 0.945 | 0.939 |
| 2011 | 0.944 | 0.940 | 0.952 | 0.943 |
| 2012 | 0.960 | 0.963 | 0.955 | 0.958 |
| 2013 | 0.926 | 0.918 | 0.933 | 0.924 |
| 2014 | 0.940 | 0.941 | 0.933 | 0.937 |
| 2015 | 0.966 | 0.964 | 0.967 | 0.966 |
| 2016 | 0.921 | 0.922 | 0.921 | 0.921 |
| 2017 | 0.924 | 0.919 | 0.925 | 0.921 |
| 2018 | 0.986 | 0.988 | 0.985 | 0.986 |
| 2019 | 0.924 | 0.919 | 0.925 | 0.921 |

## Appendix C: Interannual LST variability

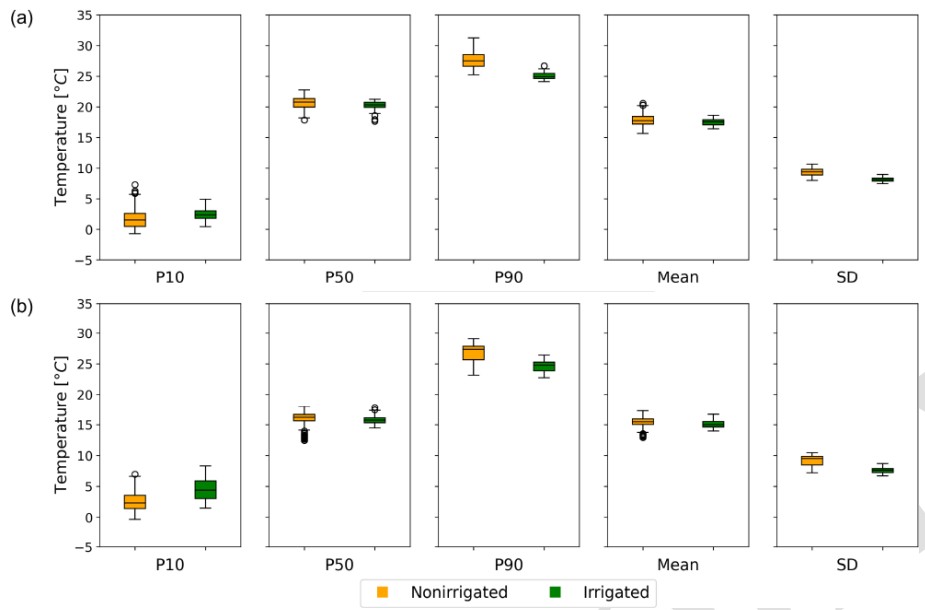

**Figure C1. (b)** The box plot shows a statistical summary of LST data for non-irrigated and irrigated pixels for **(a)** 2018 and **(b)** 2019.

*Code and data availability.* Code and data will be published in the 4TU repository. The radiation term for input of the land surface temperature module was retrieved from https://datalsasaf.lsasvcs.ipma.pt/PRODUCTS/MSG/MDIDSSF/NETCDF/ (LSA-SAF, 2024a), and the surface albedo was retrieved from https://datalsasaf.lsasvcs.ipma.pt/PRODUCTS/MSG/MDAL/NETCDF/ (LSA-SAF, 2024b). The MODIS land surface temperature data from the Terra and Aqua sensors were retrieved from https://lpdaac.usgs.gov (last access: TS2 ; DOI: https://doi.org/10.5067/MODIS/MOD11A1.061, Wan et al., 2021). The irrigation statistics at NUTS level 2 for data validation are available in the Eurostat database (https://doi.org/10.2908/ef_poirrig, Eurostat, 2018). The irrigation map (output of this research) can be accessed here: https://github.com/dvprnmsr/irrigation_paper (last access: TS3 ; DOI: https://doi.org/10.4121/66647538-ed17-4dd2-9af8-962ed0c61177.v2, Purnamasari et al., 2025 TS4 )

*Author contributions.* DP was responsible for developing the methodologies, performing the analysis, and writing the article. AJT and AHW improved the article, reviewed the figures, and refined the experimental setup. AJT and AHW supervised DP in their PhD programme.

*Competing interests.* At least one of the (co-)authors is a member of the editorial board of *Hydrology and Earth System Sciences*. The peer-review process was guided by an independent editor, and the authors also have no other competing interests to declare.

ther geographical representation in this paper. While Copernicus Publications makes every effort to include appropriate place names, the final responsibility lies with the authors.

*Acknowledgements.* We thank our research partners in the Horizon Europe STARS4Water project. We would like to thank Joost Buitink for providing forcing data and the model setup for wflow_sbm. Finally, we appreciate the valuable feedback from two anonymous reviewers.

*Financial support.* This research has been supported by the EU Horizon 2020 (grant no. 101059372). TS5

*Review statement.* This paper was edited by Alexander Gruber and reviewed by two anonymous referees.

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

**Remarks from the typesetter**

TS1    Please give an explanation of why Fig. 10 needs to be removed. We have to ask the handling editor for approval. Thanks.

TS2    Please provide date of last access.

TS3    Please provide date of last access.

TS4    Please confirm the added sentence.

TS5    Please confirm both Acknowledgements and Financial support sections.

TS6    Please check URL.

TS7    Please check and confirm the reference list entry. In the MS records, this has been moved to the category "Model code and software".

TS8    Please confirm addition.

TS9    Please confirm reference list entry.