# Peer review of "Identifying irrigated areas using land surface temperature and hydrological modelling: Application to Rhine basin"

_EGUsphere, 2024_

## Referee Comment (RC1)

**Egusphere-2024-1929**

The manuscript proposes an approach for mapping irrigated areas leveraging discrepancies between satellite-derived and modeled land surface temperature (LST) into a random forest model. The topic is surely timely and in line with the journal's aims. Nevertheless, some authors' choices are a bit unclear to me. I also believe there is room for improvement in the manuscript presentation. Please find specific comments as follows:

- Line 42. For the Ebro basin several methodologies for deriving irrigation amounts also have been published:
  https://essd.copernicus.org/articles/15/1555/2023/
  https://www.sciencedirect.com/science/article/pii/S0378377424001082
  https://hess.copernicus.org/articles/28/441/2024/;
- Line 52. Irrigation mapping through optical remote sensing is a rapidly evolving topic… maybe more recent works could be cited;
- Lines 59-78. The end of the Introduction generally presents the purpose of the study, but in this case this part is too long and detailed, more than what is required in the introduction section. Also, I believe that it could be mentioned the use of MODIS data as a source of satellite LST estimates;
- Line 81. The wflow_sbm should be briefly introduced earlier;
- Line 118. You may add a reference here;
- Lines 121 and 123. I believe the reference should be moved to the end of each sentence;
- Lines 138-141. I believe this is not needed. Please refer to the specific comment about this part;
- Line 150. The meaning of the terms of Eq (1) should be explained here (net radiation, sensible heat flux, etc.), not later in the text as in the current form;
- Figure 2 is not recalled in the main text;
- Line 181. The passage from Eq (12) to Eq (13) is not straightforward. It seems you are expressing H according to the bulk transfer equation (Monteith, 1973) and equalizing to Eq (12) to derive LST, if I am not wrong. However, this should be specified. Also, $\rho_a$ and $c_p$ are not defined.
- Lines 198-202. You are actually using remote sensing observations to derive the modeled LST to be used as a baseline. Can irrigation effects be present in such observations (e.g., lower albedo)?
- Line 218. Sentence 2 sounds as a bit redundant at this point;
- Lines 221-222. Please quantify the magnitude od data gaps due to cloud coverage (% rate);
- Line 225. Please note that the methodology proposed by Dari et al. (2021) has been implemented with vegetation indices also (https://www.mdpi.com/2073-4441/16/5/644);
- Line 227-230. If spatiotemporal features have been considered, why not applying the methodology to satellite LST directly? The authors should stress more the rationale of using a baseline approach;
- Section 2.4.2. One may argue that it is a Landsat-based irrigation mapping method;
- Lines 244-249. In this way, uncertainties associated to Landsat and land cover data are embedded in the irrigation maps produced. Have they been assessed/quantified somehow?

- Section 3.1. I believe this section does not add value to the paper. I suggest to move it to Appendix and enclose random forest performance instead in the main text, as it has surely impact on the irrigation maps developed;
- Lines 298-299. Please rephrase;
- Lines 300-306. It sounds a bit as a discussion;
- Lines 315-320. Yes, it is definitely a matter of spatial resolution;
- Lines 320-330. This part also seems to be a discussion rather than presentation of results;
- Lines 344-345. This is interesting. I also appreciate the related discussion later on. To fully understand if less rainfall actually means lower water availability for irrigation is reasonable in this case, one should have more information on the irrigation infrastructure (i.e., source of irrigation water, presence of reservoir, etc.);
- Figure 9. Can crop rotation explain the variability found in the irrigation frequency?
- Lines 351-354. This sounds again as discussion;
- Lines 393-395. This is a valuable result.
- Line 423. The study of Deines et al. (2019) is focused on the High Plains Aquifer, not on the Ebro basin;
- Lines 438. 10 years? N=10 is not clear;
- Lines 445. This is a known issue, corroborated by outcomes of several papers. Maybe some work could be cited. Also, I would say "irrigation maps" rather than "irrigation products".

---

## Author Comment (AC1)

**Response to referee comments: Anonymous Referee #1**

We appreciate the reviewer for dedicating their time and offering valuable suggestions to enhance the manuscript. Below, we address each specific comment and suggest revisions to address the concerns.

**Comment:** Line 42. For the Ebro basin several methodologies for deriving irrigation amounts also have been published: https://essd.copernicus.org/articles/15/1555/2023/ https://www.sciencedirect.com/science/article/pii/S0378377424001082 https://hess.copernicus.org/articles/28/441/2024/;

**Response:** Thank you for providing published methodologies for deriving irrigation amounts. We will discuss relevant methodologies and their application in the Ebro basin in our manuscript.

**Comment:** Line 52. Irrigation mapping through optical remote sensing is a rapidly evolving topic… maybe more recent works could be cited;

**Response:** Recent works related optical remote sensing would be cited.

**Comment:** Lines 59-78. The end of the Introduction generally presents the purpose of the study, but in this case this part is too long and detailed, more than what is required in the introduction section. Also, I believe that it could be mentioned the use of MODIS data as a source of satellite LST estimates;

**Response:** The last paragraph of the introduction will be revised.

**Comment:** Line 81. The wflow_sbm should be briefly introduced earlier;

**Response:** "wflow_sbm" will be changed to "a hydrological model".

**Comment:** Line 118. You may add a reference here;

**Response:** Reference will be added.

**Comment:** Lines 121 and 123. I believe the reference should be moved to the end of each sentence;

**Response:** The reference will be moved to the end of each sentence.

**Comment:** Lines 138-141. I believe this is not needed. Please refer to the specific comment about this part;

**Response:** We will review your suggestions and reconsider the relevance of this section to the overall focus of the manuscript.

**Comment:** Line 150. The meaning of the terms of Eq (1) should be explained here (net radiation, sensible heat flux, etc.), not later in the text as in the current form;

**Response:** The explanations of terms of Rn, LE, H, and G in Eq (1) will be explained earlier.

**Comment:** Figure 2 is not recalled in the main text;

**Response:** Figure 2 will be recalled in Section 2.3.

**Comment:** Line 181. The passage from Eq (12) to Eq (13) is not straightforward. It seems you are expressing H according to the bulk transfer equation (Monteith, 1973) and equalizing to Eq (12) to derive LST, if I am not wrong. However, this should be specified. Also, $\rho a$ and $cp$ are not defined.

**Response:** Thank you for pointing that out. Additional information will be provided before Eq (13). $\rho a$ and $cp$ will be defined.

**Comment:** Lines 198-202. You are actually using remote sensing observations to derive the modeled LST to be used as a baseline. Can irrigation effects be present in such observations (e.g., lower albedo)?

**Response:** While we acknowledge that irrigation effects may influence the observations, the attribution of latent heat flux derived from water balance model plays a more significant role in our case. To provide some perspective, the difference in albedo between the assumed irrigated and non-irrigated pixels results in a small temperature change. Throughout the growing season, this average difference on albedo is 0.00172, which has a weak effect on the Land Surface Temperature (LST), contributing to a change of approximately 0.0116 K. This effect will be mentioned in the revised version.

**Comment:** Line 218. Sentence 2 sounds as a bit redundant at this point;

**Response:** The redundant sentence in Line 218 will be removed.

**Comment:** Lines 221-222. Please quantify the magnitude of data gaps due to cloud coverage (% rate);

**Response:** We will calculate cloud cover percentage of MODIS LST products.

**Comment:** Line 225. Please note that the methodology proposed by Dari et al. (2021) has been implemented with vegetation indices also (https://www.mdpi.com/2073-4441/16/5/644);

**Response:** Thank you for providing the work by Dari et al. (2021). We will discuss how it relates and complements our current work.

**Comment:** Line 227-230. If spatiotemporal features have been considered, why not applying the methodology to satellite LST directly? The authors should stress more the rationale of using a baseline approach

**Response:** The spatial features identified between 227 and 230 are based on differences in land surface temperature (LST). The reasoning behind the chosen method is that, in humid areas, the signals from rainfed and irrigated land often overlap, making it difficult to perform continuous classification for irrigation mapping at the catchment scale. By combining LST observations with water balance data from hydrological modeling, this method helps to eliminate the influence of primary evapotranspiration driven by precipitation. As a result, the remaining temperature differences reflect evapotranspiration specifically related to irrigation. This information obtained derived from LST differences improves the accuracy of identifying irrigated areas compared to using LST and evapotranspiration ET observations alone. We will elaborate more about the rationale of using the approach in the introduction section.

**Comment:** Section 2.4.2. One may argue that it is a Landsat-based irrigation mapping method;

**Response:** We note that Landsat and land surface temperature derived from hydrological model are both used in the manual labeling. Thus, information may influence evaluation of the classifier evaluation on the manually labeled dataset. However, these manually labeled datasets mainly depend on visual cues which were not included in the random forest classification that produces the classification results.

**Comment:** Lines 244-249. In this way, uncertainties associated to Landsat and land cover data are embedded in the irrigation maps produced. Have they been assessed/quantified somehow?

**Response:** Thank you for your observation. We acknowledge that uncertainties related to cover data are inherent in the production of the irrigation maps. However, the uncertainties associated with Landsat imagery embedded in the produced irrigation maps are negligible, as the random forest classifier was trained on land surface temperature differences. In the revised version, uncertainties related to land cover will be discussed.

**Comment:** Section 3.1. I believe this section does not add value to the paper. I suggest to move it to Appendix and enclose random forest performance instead in the main text, as it has surely impact on the irrigation maps developed;

**Response:** We will evaluate this section and consider moving random forest performance to one of the main sections.

**Comment:** Lines 298-299. Please rephrase;

**Response:** Lines 298-299 will be rephrased.

**Comment:** Lines 300-306. It sounds a bit as a discussion; -

**Response:** We will investigate these lines.

**Comment:** Lines 315-320. Yes, it is definitely a matter of spatial resolution;

**Response:** We agree that this is indeed a matter of spatial resolution.

**Comment:** Lines 320-330. This part also seems to be a discussion rather than presentation of results;

**Response:** We will review the section and consider reworking it to make the distinction between the presentation of results and discussion clearer in future revisions.

**Comment:** Lines 344-345. This is interesting. I also appreciate the related discussion later on. To fully understand if less rainfall actually means lower water availability for irrigation is reasonable in this case, one should have more information on the irrigation infrastructure (i.e., source of irrigation water, presence of reservoir, etc.);

**Response:** Thank you for your suggestions. We agree that having more detailed information on the irrigation infrastructure, such as the sources of irrigation water or the presence of reservoirs, would provide complete insights. In future studies, this data could indeed offer a more comprehensive understanding of how reduced rainfall impacts water availability for irrigation. We appreciate your input and will consider this for upcoming work.

**Comment:** Figure 9. Can crop rotation explain the variability found in the irrigation frequency?

**Response:** Thank you for your suggestions. We will try to investigate if crop rotation had influence on the irrigated area for the year 2010 to 2019.

**Comment:** Lines 351-354. This sounds again as discussion;

**Response:** These lines will be evaluated.

**Comment:** Lines 393-395. This is a valuable result.

**Response:** Thank you for your feedback.

**Comment**: Line 423. The study of Deines et al. (2019) is focused on the High Plains Aquifer, not on the Ebro basin;

**Response**: "the Ebro basin" will be changed to "High Plains Aquifer".

**Comment:** Lines 438. 10 years? N=10 is not clear;

**Response:** "N=10" will be changed to "over a simulation period of ten years".

**Comment:** Lines 445. This is a known issue, corroborated by outcomes of several papers. Maybe some work could be cited. Also, I would say "irrigation maps" rather than "irrigation products"

**Response:** Outcomes of several papers will be added and "irrigation products" will be replaced with "irrigation maps".

---

## Author Comment (AC2)

**Response to referee comments: Anonymous Referee #2**

We appreciate the reviewer for dedicating time and offering valuable suggestions to improve the quality of our manuscript. Below, we provide detailed responses to each specific comment and the corresponding revisions made in the manuscript (written in blue) to address the concerns.

**Comment:** I believe the use of LST differences for irrigation mapping needs better justification. It raises the question of why not directly use satellite-based retrievals and evapotranspiration (ET) derived from the hydrological model to detect irrigation, instead of reverting to simulate LST from energy balance. In other words, existing methods for estimating actual irrigation could be used to identify irrigated areas simply by masking where irrigation is detected. For example, Olivera-Guerra et al. (2020, https://doi.org/10.1016/j.rse.2019.111627) used the coupling between an energy and water balance model to estimate irrigation, which was evaluated in both non-irrigated and irrigated fields. Although it is argued that errors in ET retrievals may hinder irrigation mapping, the errors involved in both satellite-based and modeled LST are equally significant. Additionally, the use of LST-derived products (e.g., ET, root-zone soil moisture, water stress) in estimating or detecting irrigation should be introduced and discussed in the introduction section, as shown by some studies (Droogers et al. 2010, https://doi.org/10.1016/j.agwat.2010.03.017; Olivera-Guerra et al. 2018, 2020, https://doi.org/10.1016/j.agwat.2018.06.014; Chen et al. 2018, https://doi.org/10.1016/j.rse.2017.10.030). Without this context, the use of LST is presented as the key point and the novelty in estimating irrigation. Therefore, I believe the novelty in using LST to detect irrigated areas should be well justified.

**Response:**

We appreciate the comments and suggestions. We agree that satellite-based retrievals of ET and other LST-derived products offer valuable opportunities for irrigation detection, as demonstrated by Olivera-Guerra et al. (2020). The method for estimating ET in the study relies on extreme LST values on the image representing dry and wet conditions to constrain the partitioning of the available surface energy. In arid and semi-arid regions, identifying these extreme values is less challenging than in humid regions due to the consistent moisture availability, high variability, and overlap of wet and dry periods in humid regions.

Regarding the justification for using LST differences for irrigation mapping, LST provides direct measurements that minimize the uncertainties associated with ET estimates derived from LST products. ET estimates from remote sensing models are highly divergent across products, with inconsistencies attributed to differences in input data, methodology, parameterization, and model structure (Vinukollu et al., 2011; Badgley et al., 2015; Zhu et al., 2022; Lehmann et al., 2022). Zhang et al. (2020) further elaborated on the significant divergence between ET estimates from energy balance approaches and residual water balance methods in humid regions. Although ET models capture monthly variations, they show different sensitivities to rainfall and often fail to capture the spatial patterns of ET from water balance methods, as well as the variability caused by ET peaks following heavy rainfall. It is argued that minimizing ET

errors can be achieved by ensuring proper partitioning of the water balance, constraining the magnitude of precipitation, and selecting high-quality datasets (Lehmann et al., 2022).

The estimate of ET from a hydrological model used in our study is constrained by potential evapotranspiration. In humid regions, when precipitation exceeds potential evapotranspiration, excess water tends to contribute to runoff rather than additional ET. To ensure this, we used a model that has been calibrated and validated across several discharge measurement stations with relatively dense gauge observations to perform well for rainfed conditions. This ensures that LST-derived ET estimates are constrained by potential evapotranspiration and that excess precipitation is accurately routed into runoff. The performance of the water balance model used in this study was validated against discharge measurements from various stations in the study basin, resulting in Kling-Gupta Efficiency (KGE) coefficients ranging from 0.60 to 0.90 (Imhoff et al. (2020)). This validation process confirms that excess rainfall is accurately partitioned into runoff rather than ET. We will also ensure to clearly explain why LST was chosen as the primary method for our study.

**Comment:** Another important point to deepen is the use of LST in wet condition (humid regions or wet years in the study area). It would be interesting to analyze differences in the classification of irrigated areas in dry and wet years to draw more conclusions about the use of LST in such conditions. For example, differences in LST or ET are more important in dry years (i.e., water-limited regimes) than in wet years (energy-limited regimes), particularly in dry years with the presence of fields where the crop water requirement is fully supplied to avoid water stress. Therefore, irrigated areas would be easier to detect in drier conditions, while more errors are likely in wet conditions (energy-limited regimes).

**Response:** Thank you for your suggestions. We divided the catchment area into subcatchments with different climatic conditions. The evaluation of classification results was conducted for the years 2013 and 2016, which had higher mean annual precipitation compared to the rest of the simulation period. This approach allowed us to assess the model's performance under dry and wet regions. The dry regions were classified as NUTS level 2 areas located in the Middle Rhine subcatchment. Meanwhile, the wet regions are located in Moselle, Neckar, Main, and Lower Rhine subcatchment. The results indicate that the classification performs better in these drier regions (see Figure 1). The evaluation of the classification is also influenced by the effects of large agricultural holdings in the dry regions. We will revise the current discussion and discuss the limitations of the methodology in wet regions.

[Figure]

Figure 1: The mapped irrigated area of the Rhine basin as identified through classification ($A_{i,sim}$) is compared with the total irrigated areas reported in Eurostat data at NUTS level 2 ($A_{i,obs}$) for the years (a) 2013 and (b) 2016.

$R_2$ values were calculated for the overall region (oa), dry regions (dr), and wet regions (wr). The values of the total irrigated areas [× 1000 ha] were transformed using log(A + 1) transformation.

**Comment:** According to Lines 301-304, the fact that the model trained with data from a specific year cannot be used to identify irrigated areas for the entire study period could justify the use of existing models for estimating irrigation and consequently detecting irrigated areas, rather than relying on LST differences. Comparing irrigation mapping using LST differences and ET differences should be performed for further analysis. Such analysis would allow for a more robust justification of the use of LST for irrigation mapping.

**Response:** We appreciate the comment regarding the limitations of using a model trained on data from a specific year across an entire study period. While this suggests the need for a more generalized classifier for multiyear irrigation detection, our study demonstrates that LST differences provide a valuable method in regions where rainfed and irrigated signals often overlap. At the plot scale, LST and ET may exhibit different magnitudes between irrigated and non-irrigated areas. However, mapping irrigated plots at the catchment level in our study region presents challenges due to insufficient distinct features between irrigated and non-irrigated areas during dry years, and even more so in years with adequate precipitation, when non-irrigated croplands exhibit the same LST temporal features as irrigated croplands (Figure 2). By using LST differences, evapotranspiration from precipitation estimated through water balance can be excluded, isolating only the evapotranspiration driven by irrigation. The temporal features of LST differences provides more distinct features for classification.

The main reason a model trained on data from a specific year cannot be applied across the entire study period is the dynamic nature of irrigation decisions, fallow practices, and the interannual variability in meteorological conditions, which also affect LST and ET products. A model trained on data from a single year may fail to account for these variabilities, as it uses LST and ET features or thresholds from irrigated or non-irrigated pixels to years with differing conditions. Although comparing irrigation mapping using both LST and ET differences could offer additional insights, the significant variability in ET products in humid catchments discussed in the first question would likely introduce uncertainties into the classification.

[Figure]

Figure 2: The box-plot shows statistical summary of LST data for non-irrigated and irrigated pixels for (a) 2018 and (b) 2019.

**Comment:** Lines 421-429. The limitations of LST in humid regions should be discussed. Even though decreased precipitation may lead to reductions in the extent of irrigated areas during the driest years, particularly in semi-arid regions (e.g., Afghanistan and the Ebro basin), this may not necessarily be the case in more humid regions where precipitation amounts are still substantial, such as the Rhine basin. In wet conditions, detection of irrigation using LST becomes more challenging and errors are more likely, leading to potential compensations that hamper the establishment of a clear relationship between precipitation and irrigated areas. Therefore, further evaluations should be carried out. For instance, Appendix B confirms that less precipitation leads to more irrigated areas, as detection is more easily captured by LST and more areas require irrigation.

**Response:** Based on the additional evaluation, we will include a more detailed discussion on the limitations of LST in humid regions and the possible causes of misclassification.

**Other comments**

**Comment:** Lines 40-42. I would recommend delving deeper into the irrigation detection in diverse climates, discussing the advantages of using LST in semi-arid to arid regions and the challenges in temperate to humid climates under an energy-limited regime.

**Response:** Thank you for your suggestions. We will discuss the advantages and disadvantages of using LST in semi-arid to arid regions as well as in temperate humid climates as follows:

"The use of LST as an indicator of crop health status resulting from irrigation has been widely applied in arid and semi-arid regions. Zhu et al. (2022) highlighted the effectiveness of using LST observations in crop model to quantify the evaporative cooling effects caused by changes in water and surface energy due to irrigation on maize croplands across Nebraska in the United States. Their findings demonstrate that LST provides valuable information on the impacts of irrigation on heat and water stress in crops. On basin level, Haddeland et al. (2006) investigated the impact of irrigation on the water and energy balances in the Colorado and Mekong River basins using the Variable Infiltration Capacity (VIC) hydrology model. The results show that irrigation increases latent heat flux, resulting in a reduction in surface temperature. On an annual scale, the cooling effect averaged 0.04°C across both basins, with a more significant decrease of up to 2.1°C in regions with high concentrations of irrigated croplands during peak irrigation months. Olivera-Guerra et al. (2020) used LST as complementary data in crop models to estimate irrigation water use. When crops experience water stress, land surface temperature increases due to increased sensible heat flux. By comparing the elevated LST values with the canopy temperature of well-watered fields, they were able to quantify coefficient of crop water stress (Ks).

While LST provides clear difference in irrigated and rainfed croplands in arid and semiarid regions, its effectiveness diminishes in more energy-limited conditions such as in temperate and humid climates. In regions with low surface energy availability, the use of LST is more challenging due to high moisture levels and reduced temperature variability, which complicate the separation of irrigation effects from natural variations in soil moisture and temperature (Roth et al., 2017). Zhang et al. (2022) used LST to estimate evapotranspiration from irrigation in the North China Plain, achieving higher accuracy during the winter season when precipitation is lower than in the summer months. Meanwhile, in the summer months, the effects of irrigation on land surface temperature are more difficult to detect. This is because precipitation often meets crop water requirements, making irrigation only supplemental and reducing its impact on land surface temperature changes. In such climates, complementary methods are required for accurate irrigation detection and monitoring. The more stable moisture levels and less pronounced temperature fluctuations in these regions make it difficult to differentiate between irrigated and non-irrigated areas based solely on LST. In this research, we propose integrating a water balance approach to account for evapotranspiration driven by precipitation, limiting the attribution of surface energy to water routed to runoff. This method will help to refine irrigation detection by excluding the effects of precipitation-induced evapotranspiration."

**Comment:** Lines 42-44 are not in context with irrigation retrievals in diverse climates.

**Response:** The paragraph containing lines 42-44 discusses current efforts to identify irrigation in both arid and humid regions, noting that fewer studies have been conducted in humid areas compared to arid and semi-arid regions. We understand that it may appear to be addressing existing approaches in irrigation retrievals in various climates. We will revise the paragraph to deliver a clearer message to the reader.

**Comment:** Line 63. Add references of existing approaches.

**Response:** References of existing approaches will be added.

**Comment:** Line 80. The wflow_sbm should be previously introduced.

**Response:** wflow_sbm will be previously introduced in the introduction section.

**Comment:** Line 139. PTFs?

**Response:** "PTFs" will be changed to "pedotransfer functions (PTFs)".

**Comment:** Section 2.3. Since the LST module is simply the inversion of the energy balance equations, I would recommend moving most of the equations related to the energy balance (for example, those from lines 160-179 and 186-197) to the appendix. This would give more prominence to the irrigation mapping methodology.

**Response:** Section 2.3 would be reworked so it would give more prominence to the irrigation mapping methodology.

**Comment:** Section 2.4.2. Classification based on visual detection is prone to errors and should be evaluated accordingly. Are there irrigated plots available to assess the classification?

**Response:** Due to the unavailability of multiyear irrigated plot data for our purpose, we had to rely on visual interpretation. We acknowledge the inherent errors in visual detection primarily due to its subjectivity. To address this, we complemented the visual detection methodology with thermal imagery, which captures differences in land surface temperature signatures at the plot scale with similar meteorological conditions. By combining these methods, the degree of uncertainty regarding the demarcation between irrigated and non-irrigated areas can be minimized, as boundaries are more accurately defined by land surface temperature. Although

no ground-based data on irrigated plots is available, we believe this approach reduces inaccuracies associated with using true-color imagery. The limitations of visual interpretation will be discussed in the revised version.

**Comment:** Lines 245-248. The presence of neighboring land cover types (floodplains and forests as mentioned by authors) may also influence agricultural fields. It would be interesting to evaluate their impact on both the classification of irrigated/non-irrigated areas and the LST itself.

**Response:** Initially, the classification was conducted for the entire catchment that showed a large part of the natural systems does not produce additional secondary evapotranspiration. However, some systems could tap into deeper subsurface water sources. This additional input for evapotranspiration results in more pixels being identified as irrigated. Therefore, we decided to mask out non-irrigated pixels. We will discuss this in the revised manuscript.

**Comment:** Figure 6. Reduce the range of the second y-axis to see more details in LST differences. Change this y-axis label to "Temperature difference".

**Response:** The range and label of the second y-axis will be revised.

**Comment:** Figure 7. Why negative differences are obtained in irrigated crops between observed and modelled LST? How is that related to possible misclassifications between irrigate/non-irrigated fields. Change the y-axis label to "Temperature difference".

**Response:** It affects only a small fraction of the data points, which suggests that its impact might be due to random error. Regarding the potential misclassification, these values are represented by p10, the least significant feature in the random forest classification. This means that p10 has minimal influence on the classification results. The y-axis label will be changed to "Temperature difference."

**Comment:** Figure 8. What are the reasons of the large underestimation of DE24 in 2013 and the overestimation in DE73 in 2016. The year could be added as title to each plot.

**Response:** The seemingly large underestimation of DE24 in 2013 and overestimation of DE73 in 2016 are influenced by the log scale, which may have exaggerated the reported values. The underestimation is 34 ha and the overestimation is 54 ha, both of which fall below the detection threshold of the spatial resolution. We will also add the year as a title to each plot.

**Comment:** Line 359. Recall the hectares of the estimated irrigated areas.

**Response:** The hectares of the estimated irrigated areas will be recalled in the text.

**Comment:** Figure 10. Correct the caption of the figure (a, b and c).

**Response:** The caption for Figure 10 will be corrected to "(a) The total irrigated area and (b) the annual sum of climatic variables: precipitation, evapotranspiration, and the difference for the Rhine basin for the period from 2010 to 2019. (c) Linear regression analysis is performed for each climatic variable compared to the annual irrigated area.

**Comment:** Line 375. east of the border?

**Response:** We admit there was a mistake in Line 375. It will be corrected to "east of the border".

**Comment:** Figure 11. Add the region (Lower, Middle and Rhine valley) as title of each figure and in the caption of the figure.

**Response:** The name of the region will be added to each figure and in the caption of the figure.

**Comment:** Figure 12. The period of representation per irrigation map could be add to the tittle of each figure.

Response: It will be added to each figure.

References:

Badgley, G., Fisher, J. B., Jiménez, C., Tu, K. P., & Vinukollu, R. (2015). On uncertainty in global terrestrial evapotranspiration estimates from choice of input forcing datasets. *Journal of Hydrometeorology, 16*(4), 1449-1455.

Haddeland, I., Skaugen, T., & Lettenmaier, D. P. (2006). Anthropogenic impacts on continental surface water fluxes. *Geophysical Research Letters, 33*(8), L08406.

Lehmann, F., Vishwakarma, B. D., & Bamber, J. (2022). How well are we able to close the water budget at the global scale?. *Hydrology and Earth System Sciences, 26*(1), 35-54.

Olivera-Guerra, L., Merlin, O., Er-Raki, S., Khabba, S., & Escorihuela, M. J. (2018). Estimating the water budget components of irrigated crops: Combining the FAO-56 dual crop coefficient with surface temperature and vegetation index data. *Agricultural water management, 208*, 120-131.

Roth, G., Harris, G., Gillies, M., Montgomery, J., & Wigginton, D. (2013). Water-use efficiency and productivity trends in Australian irrigated cotton: a review. *Crop and Pasture Science, 64*(12), 1033-1048.

Vinukollu, R. K., Wood, E. F., Ferguson, C. R., & Fisher, J. B. (2011). Global estimates of evapotranspiration for climate studies using multi-sensor remote sensing data: Evaluation of three process-based approaches. *Remote Sensing of Environment, 115*(3), 801-823.

Zhang, Z., Lin, A., Zhao, L., & Zhao, B. (2022). Attribution of local land surface temperature variations response to irrigation over the North China Plain. *Science of the Total Environment, 826*, 154104.

Zhu, P., & Burney, J. (2022). Untangling irrigation effects on maize water and heat stress alleviation using satellite data. Hydrology and Earth System Sciences, 26(3), 827-840.

---

## Author Response (AR1)

*We appreciate the reviewer for taking the time to provide valuable suggestions to improve the quality of our manuscript. Below, we present individual responses that detail our replies to each specific referee comment and the corresponding revisions made in the manuscript:*  *and* added text. *Additionally, any new changes in the revised manuscript are noted, and we have outlined overlapping responses to both referees in the "Collective Revisions" section.*

**Response to Anonymous Referee #1**

**Comment:** Line 42. For the Ebro basin several methodologies for deriving irrigation amounts also have been published: https://essd.copernicus.org/articles/15/1555/2023/ https://www.sciencedirect.com/science/article/pii/S0378377424001082 https://hess.copernicus.org/articles/28/441/2024/;

**Response:** *Thank you for providing published methodologies for deriving irrigation amounts. We added the works in the citation as it comprehensively covers the methodology for deriving irrigation amount in the Ebro basin.*

**Comment:** Line 52. Irrigation mapping through optical remote sensing is a rapidly evolving topic… maybe more recent works could be cited;

**Response:** *Recent works related optical remote sensing were cited. We replaced the previous citations with studies from:* *https://doi.org/10.5194/essd-13-5689-2021* *and* *https://doi.org/10.1016/j.jhydrol.2020.125356 .*

**Comment:** Lines 59-78. The end of the Introduction generally presents the purpose of the study, but in this case this part is too long and detailed, more than what is required in the introduction section. Also, I believe that it could be mentioned the use of MODIS data as a source of satellite LST estimates.

**Response:** *We shortened the last paragraph of introduction by excluding the detailed description of methodology and adding short summary of the proposed methodology.*

Line 116: "This paper investigates the potential of using a framework that combines evapotranspiration estimates from a spatially distributed hydrological model wflow_sbm (van Verseveld et al., 2024) and the MODIS LST product to detect and monitor irrigated areas  We use an additional surface energy balance terms. The spatial representativeness of the distributed hydrological model in providing evapotranspiration estimates is evaluated against another evapotranspiration product. Then, land surface temperature estimates are computed based on sensible heat flux and compared with observed land surface temperatures. To distinguish irrigated from non-irrigated areas, we use a supervised classification model trained and tested on a dataset collected from satellite observations with higher spatial resolution and apply the model to aggregated land surface temperature differences module that link evapotranspiration estimates to LST, enabling

direct comparison with satellite observations. Our research aims to address the following questions based on the outcomes of this study"

**Comment:** Line 81. The wflow_sbm should be briefly introduced earlier;

**Response:** *We briefly mentioned wflow_sbm in the first line of the paragraph.*

**Comment:** Line 118. You may add a reference here;

**Response:** *We rephrased the sentence, and citations were added.*

> Line 164: "From previous study by Imhoff et al. (2020), the choice of a 1 km spatial resolution is deemed relevant and sufficient for conducting assessments sufficient to capture hydrological processes at the river basin level given the availability of data for hydrological parameters."

**Comment:** Lines 121 and 123. I believe the reference should be moved to the end of each sentence.

**Response:** *The reference was moved to the end of each sentence.*

**Comment:** Lines 138-141. I believe this is not needed. Please refer to the specific comment about this part;

**Response:** *We reworked this part and added some information on the input data.*

> Line 183: "In humid regions, when precipitation exceeds potential evapotranspiration, excess precipitation tends to contribute to runoff rather than additional ET. To account for this process, the hydrological model needs to be calibrated and validated to perform well under rainfed conditions. Additionally, this ensures that LST-derived ET estimates are constrained by potential evapotranspiration and that excess precipitation is accurately routed into runoff. Here, we use the most recent wflow_sbm schematization and parameterization as developed for the Dutch Ministry of Infrastructure and Waterways (see the report by Buitink et al. (2023)). For more detailed information on the parameterization of the wflow_sbm model, calibration, and validation are provided in Imhoff et al. (2020) and Eilander et al. (2021). The performance of the water balance model used in this study was validated against discharge measurements from various stations in the study basin, resulting in Kling-Gupta Efficiency (KGE) coefficients ranging from 0.60 to 0.90 (Imhoff et al., 2020). It is important to note that wflow_sbm does not incorporate land management practices, such as irrigation, which could potentially lead to an underestimation or overestimation of actual evapotranspiration. ~~In this study, we conducted a brief evaluation of the spatial distribution of actual evapotranspiration estimated using the model parameters derived from PTFs by comparing it against GLEAM version 3.8a (GLEAM) (Martens et al., 2017) for the period 2010-2019. In this analysis, we resampled the actual evapotranspiration observations from GLEAM to a finer resolution, preserving details at the 1 km spatial resolution of wflow_sbm.~~"

**Comment:** Line 150. The meaning of the terms of Eq (1) should be explained here (net radiation, sensible heat flux, etc.), not later in the text as in the current form;

**Response:** *The explanations of terms of Rn, LE, H, and G in Eq (1) were explained before equations. However, we decided to move some of the equations to Appendix A to emphasize the methodology on the water and surface energy balance method.*

> Line 205: " Daily land surface temperature is derived from the sensible heat flux (H), where it is obtained by resolving the energy balance  equation, which requires the net available surface energy ($R_n$, W m$^{-2}$), latent heat flux (LE, W m$^{-2}$), and soil heat flux (G, W m$^{-2}$) at a daily temporal resolution. The energy balance of the land surface is calculated as follows:"

**Comment:** Figure 2 is not recalled in the main text;

**Response:** *Figure 2 is recalled in Section 2.3.*

**Comment:** Line 181. The passage from Eq (12) to Eq (13) is not straightforward. It seems you are expressing H according to the bulk transfer equation (Monteith, 1973) and equalizing to Eq (12) to derive LST, if I am not wrong. However, this should be specified. Also, $\rho a$ and $cp$ are not defined.

**Response:** *Thank you for pointing that out. Additional information is provided before Eq (13). $\rho a$ and $cp$ are defined as density of air and specific heat of air, respectively.*

**Comment:** Lines 198-202. You are actually using remote sensing observations to derive the modeled LST to be used as a baseline. Can irrigation effects be present in such observations (e.g., lower albedo)?

**Response:** *While we acknowledge that irrigation effects may influence the observations, the attribution of latent heat flux derived from water balance model plays a more significant role in our case. To provide some perspective, the difference in albedo between the assumed irrigated and non-irrigated pixels results in a small temperature change. We added the following explanation on Section 2.3.*

> Line 263: "The irrigation signals may present in the observations, however, the attribution of latent heat flux derived from the water balance model plays a more significant role in altering land surface temperature. The difference in albedo between the assumed irrigated and non-irrigated pixels results in a small temperature change. Throughout the growing season, this average difference on albedo is 0.00172, which has a weak effect on the Land Surface Temperature (LST), contributing to a change of approximately 0.0116 K."

**Comment:** Line 218. Sentence 2 sounds as a bit redundant at this point;

**Response:** *The redundant sentence was removed.*

"."

**Comment:** Lines 221-222. Please quantify the magnitude of data gaps due to cloud coverage (% rate);

**Response:** *Data gaps due to cloud cover has been calculated.*

Line 285: "Cloud cover is prevalent in the daily LST observations of the study area. A statistical analysis was carried out to quantify the data gaps in MODIS annual LST data cube during April to October from 2010 to 2019 caused by missing values from cloud cover. The results show a mean data gap due to cloud cover of approximately 59.3% over the 10-year period for Terra and Aqua, with high seasonal variation. Cloud cover was highest during in April (67.7%) and lowest during the peak of the growing season in July (48.4%). Due to data gaps resulting from cloud cover and sampling frequency limitations in observations, yearly irrigation identification was made feasible by aggregating cloud-free daily $\Delta T_s$ over one year. Therefore, irrigated areas in this study are defined as pixels where irrigation is detected within a given year. In cases where irrigation events are recurrent within the same year, these events are counted as a single event."

**Comment:** Line 225. Please note that the methodology proposed by Dari et al. (2021) has been implemented with vegetation indices also (https://www.mdpi.com/2073-4441/16/5/644);
**Response:** *Thank you for providing the work.*

**Comment:** Line 227-230. If spatiotemporal features have been considered, why not applying the methodology to satellite LST directly? The authors should stress more the rationale of using a baseline approach.

**Response:** *The spatial features identified between 227 and 230 are based on differences in land surface temperature (LST). The reasoning behind the chosen method is that, in humid areas, the signals from rainfed and irrigated land often overlap, making it difficult to perform continuous classification for irrigation mapping at the catchment scale. By combining LST observations with water balance data from hydrological modeling, this method helps to eliminate the influence of primary evapotranspiration driven by precipitation. As a result, the remaining temperature differences reflect evapotranspiration specifically related to irrigation. This information obtained derived from LST differences improves the accuracy of identifying irrigated areas compared to using LST and evapotranspiration ET observations alone. We elaborated more about the rationale of using the approach in the introduction section. Please refer to the "Collective Revisions" section.*

**Comment:** Section 2.4.2. One may argue that it is a Landsat-based irrigation mapping method;

**Response:** *We note that Landsat and land surface temperature derived from hydrological model are both used in the manual labeling. Thus, information may influence evaluation of the classifier evaluation on the manually labeled dataset. However, these manually labeled datasets mainly depend on visual cues which were not included in the random forest classification that produces the classification results.*

**Comment:** Lines 244-249. In this way, uncertainties associated to Landsat and land cover data are embedded in the irrigation maps produced. Have they been assessed/quantified somehow?

**Response:** *Thank you for your observation. We acknowledge that uncertainties related to cover data are inherent in the production of the irrigation maps. We compared CORINE land cover data against GlobCover 2009, which resulted in a variation with a 2.8% difference in the average total cropland area. This difference equals approximately 3100 km². We discussed this briefly in the discussion stating that error in classification from landcover propagates to the classification results. Meanwhile, the uncertainties associated with Landsat imagery embedded in the produced irrigation maps are negligible, as the random forest classifier was trained on land surface temperature differences. We added the following sentence:*

> Line 311: "The  dataset collected from this procedure are used as point labels for the classifier trained on $\Delta T_s$ data."

**Comment:** Section 3.1. I believe this section does not add value to the paper. I suggest to move it to Appendix and enclose random forest performance instead in the main text, as it has surely impact on the irrigation maps developed;

**Response:** *We reworked this section.*

**Comment:** Lines 298-299. Please rephrase;

**Response:** *Lines 298-299 was rephrased.*

> "These  distinct daily temporal patterns of $\Delta T_s$ between irrigated and non-irrigated pixels were used to estimate annual irrigation extent."

**Comment:** Lines 300-306. It sounds a bit as a discussion; -

**Response:** *We moved these sentences as part of discussion.*

**Comment:** Lines 315-320. Yes, it is definitely a matter of spatial resolution;

**Response:** *We agree that this is indeed a matter of spatial resolution.*

**Comment:** Lines 320-330. This part also seems to be a discussion rather than presentation of results;

**Response:** *We moved these sentences as part of discussion.*

**Comment:** Lines 344-345. This is interesting. I also appreciate the related discussion later on. To fully understand if less rainfall actually means lower water availability for irrigation is reasonable in this case, one should have more information on the irrigation infrastructure (i.e., source of irrigation water, presence of reservoir, etc.);

**Response:** *Thank you for your suggestions. We agree that having more detailed information on the irrigation infrastructure, such as the sources of irrigation water or the presence of reservoirs, would provide complete insights. In future studies, this data could indeed offer a more comprehensive understanding of how reduced rainfall impacts water availability for irrigation. We appreciate your input and will consider this for upcoming work.*

**Comment:** Figure 9. Can crop rotation explain the variability found in the irrigation frequency?

**Response:** *Crop rotation that particularly include fallowing may have possible impact on the variability found in the irrigation frequency throughout the study period. However, there are no specific study that provides information regarding the impact of crop rotation on irrigation in our study area.*

**Comment:** Lines 351-354. This sounds again as discussion;

**Response:** *We moved these sentences as part of discussion.*

**Comment:** Lines 393-395. This is a valuable result.

**Response:** *Thank you!*

**Comment**: Line 423. The study of Deines et al. (2019) is focused on the High Plains Aquifer, not on the Ebro basin;

**Response**: *"the Ebro basin" was changed to "High Plains Aquifer".*

**Comment:** Lines 438. 10 years? N=10 is not clear;

**Response:** *"N=10" was changed to "over a simulation period of ten years".*

**Comment:** Lines 445. This is a known issue, corroborated by outcomes of several papers. Maybe some work could be cited. Also, I would say "irrigation maps" rather than "irrigation products"

**Response:** *Outcomes of several papers were added and "irrigation products" were replaced with "irrigation maps".*

**Response to Anonymous Referee #2**

**Comment:** I believe the use of LST differences for irrigation mapping needs better justification. It raises the question of why not directly use satellite-based retrievals and evapotranspiration (ET) derived from the hydrological model to detect irrigation, instead of reverting to simulate LST from energy balance. In other words, existing methods for estimating actual irrigation could be used to identify irrigated areas simply by masking where irrigation is detected. For example, Olivera-Guerra et al. (2020, https://doi.org/10.1016/j.rse.2019.111627) used the coupling between an energy and water balance model to estimate irrigation, which was evaluated in both non-irrigated and irrigated fields. Although it is argued that errors in ET retrievals may hinder irrigation mapping, the errors involved in both satellite-based and modeled LST are equally significant. Additionally, the use of LST-derived products (e.g., ET, root-zone soil moisture, water stress) in estimating or detecting irrigation should be introduced and discussed in the introduction section, as shown by some studies (Droogers et al. 2010, https://doi.org/10.1016/j.agwat.2010.03.017; Olivera-Guerra et al. 2018, 2020, https://doi.org/10.1016/j.agwat.2018.06.014; Chen et al. 2018, https://doi.org/10.1016/j.rse.2017.10.030). Without this context, the use of LST is presented as the key point and the novelty in estimating irrigation. Therefore, I believe the novelty in using LST to detect irrigated areas should be well justified.

**Response:**

*We appreciate the comments and suggestions. We agree that satellite-based retrievals of ET and other LST-derived products offer valuable opportunities for irrigation detection, as demonstrated by Olivera-Guerra et al. (2020). The method for estimating ET in the study relies on extreme LST values on the image representing dry and wet conditions to constrain the partitioning of the available surface energy. In arid and semi-arid regions, identifying these extreme values is less challenging than in humid regions due to the consistent moisture availability, high variability, and overlap of wet and dry periods in humid regions.*

*Regarding the justification for using LST differences for irrigation mapping, LST provides direct measurements that minimize the uncertainties associated with ET estimates derived from LST products. We elaborated more about the rationale of using the approach in the introduction section. Please refer to the "Collective Revisions" section.*

**Comment:** Another important point to deepen is the use of LST in wet condition (humid regions or wet years in the study area). It would be interesting to analyze differences in the classification of irrigated areas in dry and wet years to draw more conclusions about the use of LST in such conditions. For example, differences in LST or ET are more important in dry years (i.e., water-limited regimes) than in wet years (energy-limited regimes), particularly in dry years with the presence of fields where the crop water requirement is fully supplied to avoid water stress. Therefore, irrigated areas would be easier to detect in drier conditions, while more errors are likely in wet conditions (energy-limited regimes).

**Response:** *Thank you for your suggestions. We divided the catchment area into sub-basins with different climatic conditions. This approach allowed us to assess the model's performance under dry and wet regions. The dry regions were classified as NUTS level 2 areas located in the Middle Rhine sub-basins. Meanwhile, the wet regions are in Moselle, Neckar, Main, and*

*Lower Rhine sub-basins. The results indicate that the classification performs better in these drier regions (see Figure 1).*

*The following sentences were added in Section 2.4.3:*

Line 339: "The classification results were evaluated for: i) overall, ii) dry and iii) wet NUTS2 regions which were defined based on climatology of precipitation and potential evapotranspiration summarized in Table 1. The dry regions were classified as NUTS level 2 regions that lie within the Middle Rhine sub-basins. Meanwhile, the wet regions are in Moselle, Neckar, Main, and Lower Rhine sub-basins."

*The following sentences were added in Section 3.3:*

Line 398: "The mapping methodology performs better in dry regions than in wet regions. For dry regions, the $R^2$ values are 0.9 and 0.87, while for wet regions, the $R^2$ values are 0.705 and 0.783 for 2013 and 2016, respectively."

[Figure]

Figure 1: The mapped irrigated area of the Rhine basin as identified through classification ($A_{i,sim}$) is compared with the total irrigated areas reported in Eurostat data at NUTS level 2 ($A_{i,obs}$) for the years (a) 2013 and (b) 2016. $R_2$ values were calculated for the overall region (oa), dry regions (dr), and wet regions (wr). The values of the total irrigated areas [× 1000 ha] were transformed using $\log(A + 1)$ transformation.

**Comment:** According to Lines 301-304, the fact that the model trained with data from a specific year cannot be used to identify irrigated areas for the entire study period could justify the use of existing models for estimating irrigation and consequently detecting irrigated areas, rather than relying on LST differences. Comparing irrigation mapping using LST differences and ET differences should be performed for further analysis. Such analysis would allow for a more robust justification of the use of LST for irrigation mapping.

**Response:** *We appreciate the comment regarding the limitations of using a model trained on data from a specific year across an entire study period. While this suggests the need for a more generalized classifier for multiyear irrigation detection, our study demonstrates that LST*

*differences provide a valuable method in regions where rainfed and irrigated signals often overlap. At the plot scale, LST and ET may exhibit different magnitudes between irrigated and non-irrigated areas. However, mapping irrigated plots at the catchment level in our study region presents challenges due to insufficient distinct features between irrigated and non-irrigated areas during dry years, and even more so in years with adequate precipitation, when non-irrigated croplands exhibit the same LST temporal features as irrigated croplands (Figure 2). By using LST differences, evapotranspiration from precipitation estimated through water balance can be excluded, isolating only the evapotranspiration driven by irrigation. The temporal features of LST differences provides more distinct features for classification.*

*The main reason a model trained on data from a specific year cannot be applied across the entire study period is the dynamic nature of irrigation decisions, fallow practices, and the interannual variability in meteorological conditions, which also affect LST and ET products. A model trained on data from a single year may fail to account for these variabilities, as it uses LST and ET features or thresholds from irrigated or non-irrigated pixels to years with differing conditions. Although comparing irrigation mapping using both LST and ET differences could offer additional insights, the significant variability in ET products in humid catchments discussed in the first question would likely introduce uncertainties into the classification. We added the following sentences in the discussion:*

> Line 492: "Mapping irrigated plots at the catchment level in our study region presents challenges due to insufficient distinct features between irrigated and non-irrigated areas during dry years, and even more so in years with adequate precipitation, when non-irrigated croplands exhibit the same LST temporal features as irrigated croplands Appendix C. By using $\Delta T_s$ obtained from observations and hydrological models, evapotranspiration from precipitation estimated through water balance can be excluded, isolating only the evapotranspiration driven by irrigation. In our study area, the temporal patterns of $\Delta T_s$ provide more distinctive features for classification compared to using LST alone. However, the proposed methods still face challenges related to the interannual variability of $\Delta T_s$, which results in a year-specific model. Several reasons may be due to dynamic nature of irrigation decisions, fallow practices, and the interannual variability in meteorological conditions. A model trained on data from a single year may fail to account for these variabilities, as it uses LST features or thresholds from irrigated or non-irrigated pixels to years with differing conditions."

[Figure]

Figure 2: The box-plot shows statistical summary of LST data for non-irrigated and irrigated pixels for (a) 2018 and (b) 2019.

**Comment:** Lines 421-429. The limitations of LST in humid regions should be discussed. Even though decreased precipitation may lead to reductions in the extent of irrigated areas during the driest years, particularly in semi-arid regions (e.g., Afghanistan and the Ebro basin), this may not necessarily be the case in more humid regions where precipitation amounts are still substantial, such as the Rhine basin. In wet conditions, detection of irrigation using LST becomes more challenging and errors are more likely, leading to potential compensations that hamper the establishment of a clear relationship between precipitation and irrigated areas. Therefore, further evaluations should be carried out. For instance, Appendix B confirms that less precipitation leads to more irrigated areas, as detection is more easily captured by LST and more areas require irrigation.

**Response:** *Based on the additional evaluation, we observed that irrigation detection in humid regions is more challenging due to less pronounced LST difference compared to dry years that has resulted in more error. Therefore, we moved the analysis from Appendix B into the main text because the region has the highest irrigation and discussed the challenges in identifying the irrigation area.*

*The following sentences is added to the Section 3.2:*

> Line 435: "At basin level, there is a positive correlation between annual irrigated area and precipitation. However, Figure 7 highlights challenges in irrigation identification in more humid regions. As classification performance in dry regions is higher than in more

humid conditions, we use the Alsace region as an example of how climatic factor has influence on irrigated areas as it has the highest irrigated area in the region with an average of 65,860 ha."

We revised the Lines 421-429 as follows:

Line 540: "While the proposed method performs reasonably well at the basin level, challenges remain in accurately detecting irrigated areas in humid regions, as highlighted by Zhang et al. (2022). Lower performance during wet regions can be partially explained by temporal dynamics of ΔTs that showed less variability in wet years than in dry years (Roth et al., 2013). In Alsace region where irrigation is prevalent, decreased precipitation leads to  an increase in the extent of irrigated area  areas during the driest years. This observed trend contrasts with most studies conducted in arid to semi-arid regions (e.g., Afghanistan, High Plains), which highlight the impact of limited water availability on irrigation  decision-making. Decreased irrigated areas in arid to semi-arid regions can be explained by Foster et al. (2014) who demonstrate that farmers often prioritize maintaining soil water availability to minimize the risk of significant production losses by concentrating water supply on a smaller area. This irrigation strategy is constrained by regulatory restrictions that limit water abstraction. Therefore, our finding suggests that famers increasingly rely on irrigation during periods of reduced precipitation to mitigate the risk of yield loss. This highlights the need to further evaluate how much pressure from irrigation water use on water availability during drought. Although other factors influencing irrigation dynamics, such as improvements in irrigation efficiency, regulations, and restrictions on groundwater, were not studied, they may significantly influence the temporal dynamics of irrigation and needs to be investigated".

**Other comments from Anonymous Referee #2**

**Comment:** Lines 40-42. I would recommend delving deeper into the irrigation detection in diverse climates, discussing the advantages of using LST in semi-arid to arid regions and the challenges in temperate to humid climates under an energy-limited regime.

**Response:** *Thank you for your suggestions. We discussed the advantages and disadvantages of using LST in semi-arid to arid regions as well as in temperate humid climates as written in the "Collective Revisions".*

**Comment:** Lines 42-44 are not in context with irrigation retrievals in diverse climates.

**Response:** *The paragraph containing lines 42-44 discusses current efforts to identify irrigation in both arid and humid regions, noting that fewer studies have been conducted in humid areas compared to arid and semi-arid regions. We understand that it may appear to be addressing existing approaches in irrigation retrievals in various climates.*

*We revised the paragraph as follows:*

Line 58: "…These indices typically capture vegetation health and growth stages, with irrigated fields exhibiting higher values  than adjacent non-irrigated fields.

However, most studies are performed in areas with negligible precipitation during the growing season, where spectral difference is more pronounced. In temperate regions, distinguishing between irrigated and non-irrigated croplands using vegetation indices  is challenging as irrigation often supplements precipitation, which leads to overlap in the spectral signatures of irrigated and non-irrigated pixels. A study by Shamal and Weatherhead (2014) revealed that the spectral signatures between irrigated and non-irrigated croplands in the UK were identical because non-irrigated croplands experienced less water stress owing to regular precipitation. Similar finding from Ozdogan and Gutman (2008) who attempted to identify irrigated areas in the US, but the performance results deteriorated when applied to the humid eastern regions…"

**Comment:** Line 63. Add references of existing approaches.

**Response:** *References of existing approaches: https://doi.org/10.1038/s41598-017-06359-w and https://doi.org/10.14358/PERS.78.8.861 were added.*

**Comment:** Line 80. The wflow_sbm should be previously introduced.

**Response:** *wflow_sbm is briefly mentioned in the last paragraph of introduction.*

**Comment:** Line 139. PTFs?

**Response:** *"PTFs" were changed to "pedotransfer functions (PTFs)".*

**Comment:** Section 2.3. Since the LST module is simply the inversion of the energy balance equations, I would recommend moving most of the equations related to the energy balance (for example, those from lines 160-179 and 186-197) to the appendix. This would give more prominence to the irrigation mapping methodology.

**Response:** *Section 2.3 was reworked to emphasize on calculation of LST from the energy balance.*

**Comment:** Section 2.4.2. Classification based on visual detection is prone to errors and should be evaluated accordingly. Are there irrigated plots available to assess the classification?

**Response:** *We discussed the limitations of the visual interpretation in the classification presented in Section 2.4.2 as follows:*

Line 305: "Due to the unavailability of multiyear observations data for our purpose, we had to rely on true and thermal imagery with high spatial resolution to collect point data. To minimize errors in visual detection due to its subjectivity, we complemented the visual detection with thermal imagery that captures differences in land surface temperature signatures at the plot scale with similar meteorological conditions. Combining these methods can reduce the degree of uncertainty regarding the

demarcation between irrigated and non-irrigated areas due to additional information provided by land surface temperature."

**Comment:** Lines 245-248. The presence of neighboring land cover types (floodplains and forests as mentioned by authors) may also influence agricultural fields. It would be interesting to evaluate their impact on both the classification of irrigated/non-irrigated areas and the LST itself.

**Response:** *Initially, the classification was conducted for the entire catchment, which showed that a large portion of non-croplands does not show a large temperature difference. However, neighboring land cover types such as floodplains are influenced by lateral inflow while forests have deep root systems that can access deeper groundwater sources that leads to additional evapotranspiration (van Dijk et al., 2015). By including these data in the training data that were labelled as "non-irrigated", it added complexities and confusion to the random forest classifier that leads to inconclusive results.*

**Comment:** Figure 6. Reduce the range of the second y-axis to see more details in LST differences. Change this y-axis label to "Temperature difference".

**Response:** *The range was reduced and the label of the second y-axis were change to "ΔTs" that denotes temperature difference.*

**Comment:** Figure 7. Why negative differences are obtained in irrigated crops between observed and modelled LST? How is that related to possible misclassifications between irrigate/non-irrigated fields. Change the y-axis label to "Temperature difference".

**Response:** *We explained why negative differences are obtained in temperature difference in the revised manuscript. The y-axis label was changed to "ΔTs" that denotes temperature difference.*

> "Small fractions of data points with negative LST difference due to random error in Figure 5 are represented by p10. It has minimal influence on the classification results."

**Comment:** Figure 8. What are the reasons of the large underestimation of DE24 in 2013 and the overestimation in DE73 in 2016. The year could be added as title to each plot.

**Response:** *The year as title to each plot was added and the reasons are explained as follows:*

> Line 403: *"The seemingly large underestimation of Oberfranken (DE24) in 2013 and overestimation of Kassel (DE73) in 2016 are influenced by the logarithmic scale, which exaggerates the reported and predicted values. The underestimation is ~34 ha and the overestimation is ~54 ha, both of which fall below the detection threshold of spatial resolution."*

**Comment:** Line 359. Recall the hectares of the estimated irrigated areas.

**Response:** *The hectares of the estimated irrigated areas is recalled in the text.*

"Our estimated irrigated area, which averages 159 thousand hectares, exceeds the actual irrigated area (AEI) reported by GMIA (148 thousand hectares) …"

**Comment:** Figure 10. Correct the caption of the figure (a, b and c).

**Response:** *The caption for Figure 10 was corrected.*

"(a) The total irrigated area and (b) the annual sum of climatic variables: precipitation, evapotranspiration, and the difference for the Rhine basin for the period from 2010 to 2019. (b) (c) Linear regression analysis is performed for each climatic variable compared to the annual irrigated area".

**Comment:** Line 375. east of the border?

**Response:** *We admit there was a mistake in Line 375. It was change to "east of the border".*

**Comment:** Figure 11. Add the region (Lower, Middle and Rhine valley) as title of each figure and in the caption of the figure.

**Response:** *The name of the region was added to each figure and in the caption of the figure.*

**Comment:** Figure 12. The period of representation per irrigation map could be add to the tittle of each figure.

**Response**: *The period of representation was added to each figure.*

**Collective Revisions**

We shortened the introduction to add these following paragraphs.

- Rationale using LST and LST difference to detect irrigated areas:

*The following sentences were added in Introduction section to explain why LST is chosen as the primary method for our study:*

[revised manuscript text omitted]

Badgley, G., Fisher, J. B., Jiménez, C., Tu, K. P., & Vinukollu, R. (2015). On uncertainty in global terrestrial evapotranspiration estimates from choice of input forcing datasets. *Journal of Hydrometeorology, 16*(4), 1449-1455.

Haddeland, I., Skaugen, T., & Lettenmaier, D. P. (2006). Anthropogenic impacts on continental surface water fluxes. *Geophysical Research Letters, 33*(8), L08406.

Lehmann, F., Vishwakarma, B. D., & Bamber, J. (2022). How well are we able to close the water budget at the global scale?. *Hydrology and Earth System Sciences, 26*(1), 35-54.

Olivera-Guerra, L., Merlin, O., Er-Raki, S., Khabba, S., & Escorihuela, M. J. (2018). Estimating the water budget components of irrigated crops: Combining the FAO-56 dual crop coefficient with surface temperature and vegetation index data. *Agricultural water management, 208*, 120-131.

Roth, G., Harris, G., Gillies, M., Montgomery, J., & Wigginton, D. (2013). Water-use efficiency and productivity trends in Australian irrigated cotton: a review. *Crop and Pasture Science, 64*(12), 1033-1048.

Vinukollu, R. K., Wood, E. F., Ferguson, C. R., & Fisher, J. B. (2011). Global estimates of evapotranspiration for climate studies using multi-sensor remote sensing data: Evaluation of three process-based approaches. *Remote Sensing of Environment, 115*(3), 801-823.

Zhang, Z., Lin, A., Zhao, L., & Zhao, B. (2022). Attribution of local land surface temperature variations response to irrigation over the North China Plain. *Science of the Total Environment, 826*, 154104.

Zhu, P., & Burney, J. (2022). Untangling irrigation effects on maize water and heat stress alleviation using satellite data. Hydrology and Earth System Sciences, 26(3), 827-840.